# Impact of Isolated Atmospheric Aging processes on the Cloud Condensation Nuclei-activation of Soot Particles

Franz Friebel[1], Prem Lobo[1, 2, a], David Neubauer[1], Ulrike Lohmann[1], Saskia Drossaart van Dusseldorp[1], Evelyn Mühlhofer[1], Amewu A. Mensah[1,b]

[1]Institute for Atmospheric and Climate Science, ETH Zurich, Zurich, 8092, Switzerland
[2]Center of Excellence for Aerospace Particulate Emissions Reduction Research, Missouri University of Science and Technology, Rolla, Missouri 65401, USA
[a]now at: Metrology Research Centre, National Research Council Canada, Ottawa, Ontario K1A 0R6, Canada
[b]now at: Environmental and Health Protection Service, City of Zurich, Zurich, 8003, Switzerland

*Correspondence to*: Franz Friebel (franz.friebel@env.ethz.ch) and Amewu A. Mensah (a.mensah@alumni.ethz.ch)

**Abstract.** The largest contributors to the uncertainty in assessing the anthropogenic contribution in radiative forcing are the direct and indirect effects of aerosol particles on the Earth's radiative budget. Soot particles are of special interest since their properties can change significantly due to aging processes once they are emitted to the atmosphere. Probably the largest obstacle for the investigation of these processes in the laboratory is the long atmospheric lifetime of one week, demanding tailored experiments that cover this time span. This work presents results on the ability of two types of soot, obtained using a miniCAST soot generator, to act as cloud condensation nuclei (CCN) after exposure to atmospherically relevant levels of ozone ($O_3$) and humidity. Aging times of up to 12 h were achieved by successful application of the continuous-flow stirred tank reactor (CSTR) concept while allowing for size-selection of particles prior to the aging step. Particles of 100 nm diameter and rich in organic carbon (OC) that were initially CCN-inactive showed significant CCN-activity at supersaturations (SS) down to 0.3 % after 10 h of exposure to 200 ppb of $O_3$. While this process was not affected by different levels of relative humidity in the range of 5 -75 %, a high sensitivity towards the ambient/reaction temperature was observed. Soot particles with a lower OC-content demanded an approximately four-fold longer aging duration to show CCN-activity at the same SS. Prior to the slow change in the CCN-activity, a rapid increase in the particle diameter was detected which occurred within several minutes. This study highlights the applicability of the CSTR approach for the simulation of atmospheric aging processes, as aging durations beyond 12 h can be achieved in comparably small aerosol chamber volumes (<3 m$^3$). Implementation of our measurement results into a global aerosol-climate model ECHAM6.3-HAM2.3 showed a statistically significant increase in the regional and global CCN burden and cloud droplet number concentration.

## 1    Introduction

Aerosols are defined as fine solid particles or liquid droplets suspended in a gas phase. Aerosol particles impact the Earth's radiative budget both directly (e.g. through scattering of shortwave and absorption of shortwave and longwave radiation) (Haywood and Boucher, 2000) and indirectly (e.g. by changing the properties of clouds; Ackerman et al., 2000; Lohmann and Feichter, 2005; Seinfeld et al., 2016; Twomey, 1977). Furthermore, they demonstrate significant impacts on air quality and human health (Anenberg et al., 2012; Janssen et al., 2011). The chemical and physical properties of atmospheric aerosol particles are highly variable and depend on surface (land vs. ocean), region (urban vs. remote), source (anthropogenic vs. biogenic), and many more aspects. Additionally, ambient aerosol particles undergo physicochemical modification processes throughout their atmospheric lifetime (Monks et al., 2009). Condensation of gas phase volatile material or heterogeneous oxidation are general examples of these processes that are referred to as aging. An example of the multiple ways these processes modify the physicochemical properties of the particles is the change in the water affinity of an initially hydrophobic particle.

Accumulation of hygroscopic material on the surface can cause such a particle to become hydrophilic (Dalirian et al., 2018; Henning et al., 2012; Khalizov et al., 2009). Depending on the surrounding conditions, the particle can accumulate water vapor and form a droplet as in a cloud or in fog.

The process of forming a cloud droplet is called cloud droplet activation and the respective particles are called Cloud Condensation Nuclei (CCN). The particles' ability to act as CCN depends on its properties, such as size, morphology, and chemical composition (Köhler, 1936; Sorjamaa and Laaksonen, 2007). While particles consisting of hygroscopic compounds such as sea salt have a high CCN-activity, other particles such as soot (also referred to as black carbon, BC) show extremely low CCN-activity (Petzold et al., 2013). Nevertheless, BC particles have been found in cloud droplet and ice crystal residuals in ambient measurements, indicating that within their atmospheric lifetime these particles are incorporated into hydrometeors (Cozic et al., 2008; Hiranuma et al., 2013).

Soot is a by-product of incomplete combustion. Depending on its origin, soot varies greatly in chemical composition, size, and co-emitted substances. Soot particles have an atmospheric lifetime of up to one week, which is long compared to other aerosol particles (Textor et al., 2006). Currently, the impact of soot particles on human health, environment, and climate is of scientific and economic interest. Understanding their carcinogenic nature (WHO, 2016) or their impact on crops (Burney and Ramanathan, 2014) are only some examples of the increased interest in soot particles during the last few decades. Regarding their atmospheric impact, a good understanding has been gained with respect to their direct effect on visibility and air quality but their indirect climate impact, i.e. on clouds and cloud formation, remains highly uncertain (IPCC, 2013).

The pathway of how soot particles end up in hydrometeor residuals remains a major topic of discussion. On the one hand, these hydrophobic particles show little interaction with water and are reported to be poor CCN and ice-nucleating particles, respectively (INPs; Friedman et al., 2011; Koehler et al., 2009; Kulkarni et al., 2016). On the other hand, field measurements show that soot particles are enriched in cloud droplets and ice crystals compared to interstitial particles (Cozic et al., 2008; Hiranuma et al., 2013). These findings indicate that soot particles can become incorporated into hydrometeors beyond impaction scavenging potentially by an increase in hygroscopicity upon atmospheric aging turning them into CCN or INPs, respectively. The details and relevance of atmospheric aging processes potentially causing such a significant change in CCN-activity of soot particles are not well understood yet. Besides the complexity of aerosol particles, one of the challenges lies in investigating these processes in the laboratory at atmospheric conditions. Furthermore, modeling studies show that even though soot particles are poor INPs compared to dust particles (Kanji et al., 2017) they are still relevant INPs in the atmosphere due to their high abundance (Hoose et al., 2010; Savre and Ekman, 2015).

While there is a broad consensus that coating with hygroscopic substances e.g. sulfuric acid (Dalirian et al., 2018; Henning et al., 2012; Khalizov et al., 2009) increases the particle-water interaction of soot particles, the influence of oxidation processes is less well understood. The impact of oxidation processes can be investigated by simulating atmospheric aging under controlled laboratory conditions. One of the experimental challenges is to achieve extended aging time periods because the average atmospheric lifetime of soot particles is approximately one week (Textor et al., 2006). A common approach is the application of (photo-) Oxidation Flow Reactors (OFR) like the Potential Aerosol Mass (PAM) chamber (Kang et al., 2007), the Toronto Photo-Oxidation Tube (TPOT; George et al., 2007), the Micro Smog Chamber (MSC; Keller and Burtscher, 2012), the TUT Secondary Aerosol Reactor (TSAR; Simonen et al., 2017) or the Photochemical Emission Aging flow tube Reactor (PEAR; Ihalainen et al., 2019). Within these devices, the residence time ranges from 3 to 170 s and the OH-radical concentration ranges from $4.9 \times 10^8$ to $130 \times 10^8$ molec·cm$^{-3}$, while the average atmospheric OH-radical concentration is orders of magnitude lower with $1.5 \times 10^6$ molec·cm$^{-3}$ (Mao et al., 2009). The exposure conditions are recalculated to an equivalent atmospheric aging time of 0.4 to 10 days (Lambe et al., 2015). This approach implies that the oxidation speed is linearly dependent on the concentration of OH-radicals which is supported by the findings of Bedjanian et al. (2010). Deploying larger aerosol chambers with several cubic meters of volume operated in batch mode allows for longer experimental durations at more atmospherically relevant oxidant concentration levels. For example, Wittbom et al., (2014) achieved aging

durations of up to 5 h in a 6 $m^3$ aerosol chamber at OH-radical concentrations ranging from $1 \times 10^6$ to $2 \times 10^6$ molec·$cm^{-3}$ which is approximately equal to one day of atmospheric aging. Both, the OFRs and the batch-aerosol chamber methods show that equivalent atmospheric aging time spans of several hours to days are required to make soot particles CCN-active at atmospherically relevant super saturations (SS) of below 0.8 % (Pruppacher and Klett, 2010).

Another very important atmospheric oxidant is ozone ($O_3$). The effect of $O_3$ oxidation on the CCN-activity of soot particles has been investigated extensively in various laboratory studies. Despite these efforts, no CCN activation at atmospheric $O_3$ concentration and atmospherically relevant SS has been reported to the authors' knowledge. However, in the range from 1200 ppb to 20,000 ppb $O_3$ a significant increase in CCN-activity of soot particles was reported for SS above 0.8 % for exposure times between 100 s and 2 h, which should correspond to atmospheric aging times of up to 3.5 days (Friedman et al.,

2011; Grimonprez et al., 2018; Lambe et al., 2015). Contrary to the agreement regarding the linearity of aging and OH-exposure within the scientific community (Renbaum and Smith, 2011), there is no such consensus concerning aging and $O_3$ exposure. While the results of the studies mentioned above are interpreted on the assumption that the oxidation speed is directly proportional to the $O_3$ concentration, Kotzick et al., (1997) reported that no impact of concentration could be detected in the range from 25 ppb to 1000 ppb $O_3$.

Studies focusing on the uptake of $O_3$ by soot particles suggest that the reaction might not follow first-order kinetics with respect to the $O_3$ gas-phase concentration e.g. (Ammann et al., 2003; Kamm et al., 1999; Lelievre et al., 2004; McCabe and Abbatt, 2009; Zelenay et al., 2011). Similar results have been found for the decomposition of Polycyclic Aromatic Hydrocarbon (PAHs) and other organic compounds condensed on aerosol particle surfaces (Bedjanian and Nguyen, 2010; McNeill et al., 2007; Pöschl et al., 2001; Shiraiwa et al., 2011). These findings combined with soot particles found in hydrometeor residuals

question the validity of extrapolations from non-atmospheric reaction conditions being used as the basis to infer atmospheric implications. To evaluate the atmospheric relevance of $O_3$ oxidation on the CCN-activity of soot particles and the impact of the $O_3$ concentration in the atmospherically relevant range, laboratory experiments should preferably be performed at atmospherically relevant conditions with respect to oxidant concentration, relative humidity (*RH*), particle number concentration as well as reaction time.

Different experimental setups provide different advantages with respect to mimicking atmospheric aging processes. OFRs have the advantage of operating at particle number concentrations prevalent in the atmosphere. Therefore, the aerosol particles can be size-selected before entering the chamber, and changes in the CCN-activity can be investigated excluding artifacts from potential size-related effects. However, because those chambers are operated at oxidant conditions that are up to 4 orders of magnitude higher than in the atmosphere (Bruns et al., 2015) and might follow atmospherically non-relevant reaction pathways

(McNeill et al., 2007). In contrast to the OFR`s, there are large aerosol chambers with several $m^3$ of volume in which aerosols can be exposed to atmospherically relevant concentrations of oxidants and trace gases (Cocker et al., 2001; Leskinen et al., 2015; Nordin et al., 2013; Paulsen et al., 2005; Platt et al., 2013; Presto et al., 2005; Rohrer et al., 2005). Because those chambers are typically operated in batch-mode, they require elevated particle number concentrations in the input flow in order to reach the desired aerosol concentration inside the chamber within a reasonable time. Therefore, these types of chambers are

often filled with non-size-selected aerosol particles, hampering the separation of CCN activation due to chemical transformation from potential particle size effects.

In this paper, we present the results from a study investigating the effect of heterogeneous $O_3$ oxidation at atmospherically relevant conditions on the CCN-activity of soot particles derived from a co-flow propane diffusion flame. The experiments

were performed within a ~3 $m^3$ stainless steel aerosol chamber operated in continuous-flow stirred tank reactor (CSTR) mode. This approach allowed us to 1) achieve aging durations of up to 12 h, 2) utilize atmospherically relevant $O_3$ concentrations of up to 200 ppb and varying levels of humidity, and 3) execute the experiments with 100 nm size-selected soot particles. The experimental results were then implemented into the global climate model ECHAM6.3-HAM2.3 (Neubauer et al., 2019; Tegen

et al., 2019). Based on these results, we discuss the impact of aging processes on the change in CCN burden and cloud droplet number concentration (CDNC) on the global as well as regional scale.

## 2    Experimental Setup

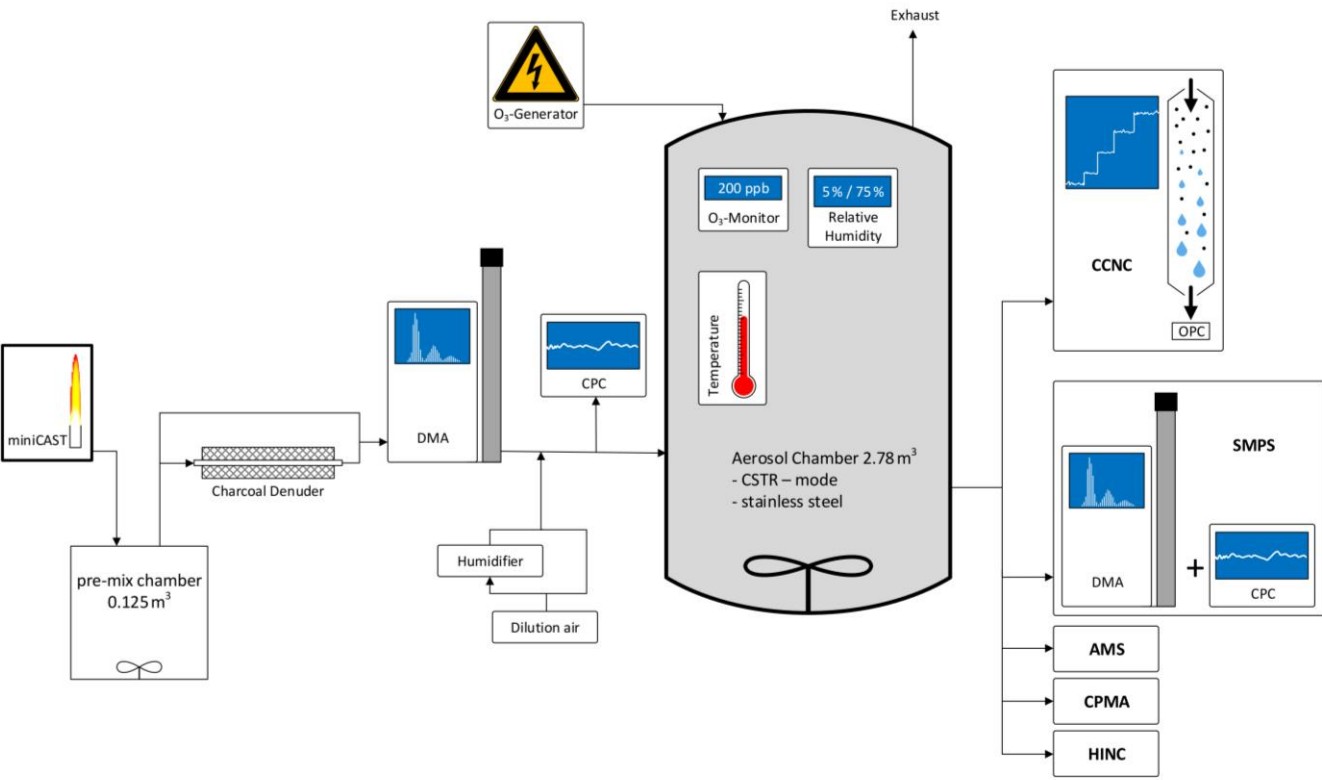

**Figure 1: Schematic of the experimental setup. Indicated are aerosol production and conditioning (left part of the figure), the aging chamber (center), and the characterization instruments (right).**

### 2.1    Overview

Multiple sets of experiments were performed in the laboratory using a 2.78 m$^3$ stainless steel aerosol chamber at ETH Zurich, Switzerland during summer 2016 and summer 2018. To allow for extended aging durations of up to 12 h, the aerosol chamber
was operated following the concept of the CSTR (Levenspiel, 1999). In accordance with this concept, soot particles were continuously added to the aerosol chamber inlet, simultaneous to the continuous withdrawal of aerosol samples from the chamber outlet for analysis. All sample lines used within the set-up consisted of stainless steel with an inner diameter of 4 mm. The sample lines were configured such that the soot aerosol could be sampled either directly upstream, i.e. bypassing, or downstream of the aerosol chamber. Friebel and Mensah (2019) introduced a new analysis technique that allows for the
retrieval of CCN activation data from experiments executed in aerosol chambers operated in CSTR mode. In addition to the investigation of the CCN-activity, data were recorded for the investigation of a broad range of physical and chemical parameters, e.g. the size and mass distribution of the particles, the INP potential, the chemical composition, and the lung deposited surface area (LDSA). Further details on the experimental set-up, instrumentation, and specific settings beyond the information given in the following section can be found in the appendix.

Soot particles were produced by a propane-fueled Jing Ltd., miniature Combustion Aerosol STandard (miniCAST 4200) generator. Such type of burners and specifically the miniCAST burner have been used widely for the production of soot particles in laboratory studies e.g. (Durdina et al., 2016; Kim et al., 2015; Malmborg et al., 2018; Mamakos et al., 2013; Maricq, 2014; Mueller et al., 2015; Török et al., 2018). The miniCAST was operated at two different settings for the generation of soot

samples with different organic carbon (OC) contents. The first sample, hereafter referred to as CAST brown (CBW), was generated under fuel-rich conditions (fuel-air ratio; FAR = 1.03). The second sample, hereafter referred to as CAST black (CBK), was generated under fuel-lean condition (FAR = 0.95) . Further details on the miniCAST set-points used during the study are listed in appendix Table 4.

The miniCAST has been used as surrogate for soot emissions from vehicle internal combustion engines (Maricq, 2014; Moore et al., 2014; Mueller et al., 2015) and aircraft engines (Bescond et al., 2014; Durdina et al., 2016). According to Marhaba et al., (2019) high engine thrust levels can be mimicked by CBK soot, while CBW soot better represents engine emissions at lower thrust levels .

The gases used were nitrogen ($N_2$) of grade 6.0 for mixing and quenching, and in-house filtered and compressed air for oxidation and dilution. The compressed air was purified by passing through a particle filter resulting in a particle concentration below the detection limit of the particle counting instrumentation. The air was further passed through a charcoal filter for the removal of volatile organic compounds (VOC). The remaining VOC content was tested by mixing the filtered air with 200 ppb of $O_3$. As no new particle formation could be detected we considered the filtered air particle- and VOC-free with respect to

our instrumentation.

After starting the burner, it was run for at least 2 h before the operating conditions were considered stable and the output was sampled. The output of the miniCAST burner was diluted by a factor of 10 using a Palas VKL 10 diluter. 6 L·min$^{-1}$ of the diluted sample was introduced into a pre-mix chamber of 0.125 m$^3$ volume. The stainless steel pre-mix chamber was air-tight and equipped with a continuously stirring fan. To allow for the selection of 100 nm particles of sufficient concentration, the

20 particles were allowed to agglomerate within the pre-mix chamber. After an average residence time of 21 min, the mode diameter of CBW and CBK particles was 90 nm and 150 nm, respectively.

### 2.2 Aerosol chamber

A 2.78 m$^3$ stainless steel aerosol chamber was used as a reaction vessel. As a detailed description of the stainless steel aerosol chamber has been previously presented by Kanji et al., (2013), only a brief description follows. The aerosol chamber is

25 equipped with a pitched blade fan of 30 cm diameter at its bottom. The fan was operated at 1000 rpm to ensure a homogenous distribution of the aerosol throughout the aerosol chamber. Based on the operational experience acquired in the summer 2016, the aluminum fan was gold plated prior to the campaign in summer 2018 to increase its conductivity and thereby reduce particle loss on its surface. Temperature, pressure, and *RH* inside the aerosol chamber were monitored by sensors mounted on a diagonally oriented taut wire. While pressure and *RH* were controlled by the conditions of the input flow, the double-wall

design of the aerosol chamber allowed for direct temperature control, which was utilized in some of the experiments. Soot aerosol and $O_3$ were introduced through individual ports on stainless steel aerosol chamber. Another port was used for the withdrawal of sample aerosol, which then was distributed to various measurement instruments.

In general, the total volumetric flow through the aerosol chamber while filling and steady state was set to 23 L·min$^{-1}$. In some experiments, a reduced flow rate of 13 L·min$^{-1}$ was applied during the overnight flushing regime to maximize the duration of

35 particle concentration above the detection limit. The experimental conditions allowed the exposure time of the soot aerosol during oxidation and humidification experiments to be extended up to 12 h.

### 2.3 Sample selection and conditioning

The soot aerosol was conditioned in multiple ways prior to entering as well as within the aerosol chamber. Following the pre-mix chamber, a home-built charcoal denuder was placed in-line for the removal of remaining gas-phase VOCs from the

40 combustion process within the miniCAST burner. The denuder consisted of a glass tube of 40 cm in length and 10 cm in diameter filled with approximately 0.7 kg of activated charcoal (Sigma-Aldrich). A metal mesh of 1.5 cm diameter connecting

the inlet and outlet of the denuder allowed the aerosol stream to pass through the center of the denuder without direct exposure to the charcoal. The denuder was bypassed (i.e. the sample was not denuded) for some of the experiments to evaluate the potential impact of the remaining VOCs on the CCN-activity of the particles. A TSI 3081L Differential Mobility Analyzer (DMA) was used downstream of the denuder for the selection of 100 nm soot particles. The DMA was operated with a sample airflow rate of 1.7 to 1.9 $L \cdot min^{-1}$ and a sheath air flow rate of 10 $L \cdot min^{-1}$. After diluting the sample airflow with 21 $L \cdot min^{-1}$ of particle- and VOC-free compressed air, a particle concentration of ~1200 $cm^{-3}$ was achieved. A TSI 3772 Condensation Particle Counter (CPC) was used to monitor the number concentration of the soot aerosol particles entering the aerosol chamber. The input concentration remained stable for the entire duration of an experimental run.

Conditioning of the soot particles with $O_3$ or elevated humidity took place within the aerosol chamber. Gas streams containing $O_3$ and water vapor were fed into the aerosol chamber through individual ports. $O_3$ was produced by a continuously running corona discharge $O_3$ tube operating on high purity 5.6 synthetic air. The output of the $O_3$ generator was diluted by a factor of 100 using a Palas VKL 100 diluter with particle- and VOC-free in-house compressed air. The flowrate of the $O_3$ stream into the stainless steel aerosol chamber was maintained at 0.040 - 0.070 $L \cdot min^{-1}$. The $O_3$ concentration within the aerosol chamber was monitored by an Aeroqual series 940 transmitter (0 - 0.5 ppm) mounted on an additional port. For sample humidification, particle- and VOC-free compressed air was split into multiple streams. One stream was passed through a silica gel diffusion dryer resulting in a *RH* of less than 5 %. The second stream was split and led through two Nafion-humidifier coil tubes surrounded by thermostated water resulting in a *RH* of up to 95 %. The temperature of the water was controlled by using an Ecoline Immersion thermostat E300 with Stainless Steel bath 006. The flow rates of the dry and the humidified air streams were regulated by individual mass flow controllers (MKS, 0 - 20 $L \cdot min^{-1}$) and mixed within a 5 L glass volume. This set-up allowed for the stable production of air at a pre-set *RH* level and a flow rate of 20 $L \cdot min^{-1}$. In addition to the sensors monitoring the humidity within the aerosol chamber, a Vaisala HMT337 humidity sensor was used to monitor the humidification air before entering the aerosol chamber.

### 2.4    Sample characterization

A suite of instruments was deployed for the characterization of the soot particle samples. Besides stationary centerpieces for the determination of the particle size distribution and CCN-activity, the specific configuration of instruments varied depending on availability. Data on the chemical composition, the INP activity, and the single-particle mass distribution was acquired in many but not all experiments.

A TSI Scanning Mobility Particle Sizer (SMPS) consisting of a TSI 3081L DMA and a TSI 3772 CPC was used to record the particle size distributions in the range of 8 – 280 nm at a scanning frequency of 135 s. The DMA was operated with a sample flow rate of 1 $L \cdot min^{-1}$ and a sheath air flow rate of 10 $L \cdot min^{-1}$. The CCN-activity of the soot particles was determined using a continuous flow Cloud Condensation Nuclei Counter (CCNC) from Droplet Measurement Technologies (DMT; Roberts and Nenes, 2005). As size-selected particles were investigated, the CCN-activity was investigated by exclusively modulating the SS in the range from 0.2 % to 1.4 %. An additional data point at a SS of 1.6 % was acquired in CBK experiments. With an acquisition duration of 6 - 10 min at each set-point, the sampling interval for a full scan was approximately 66 min.

## 2.5 Experimental procedure and experimental conditions

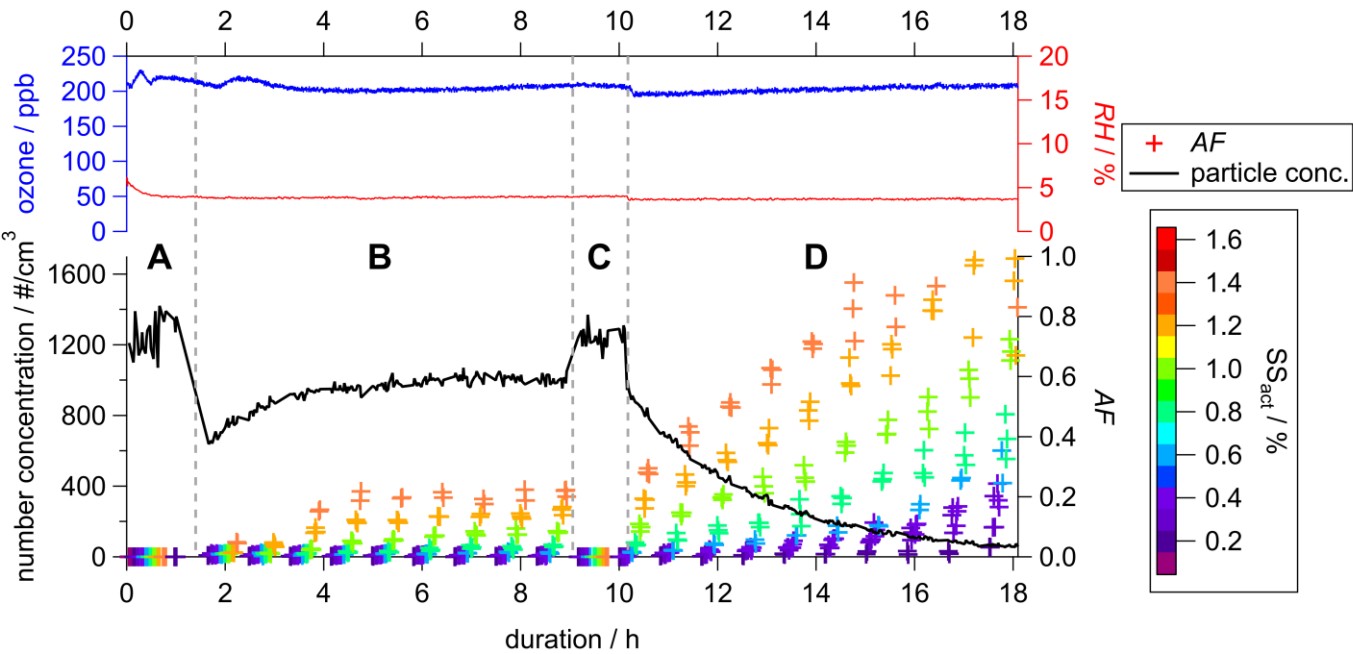

**Figure 2 Example of data of the CCN-activity (colored crosses, right bottom axis), particle number concentration (black line, left bottom axis), O₃ concentration (blue line, left top axis), and *RH* (red line, right top axis) as a function of experimental duration (bottom axis). The crosses presenting the CCN activated fraction (*AF*) are color-coded by the SS as indicated in the color scale on the right. The labels: A (bypass), B (filling), C (bypass), and D (flushing) indicate specific periods within the experiment.**

The experimental procedure and therefore the data that can be acquired differs from experiments conducted in OFRs and batch reactors. As an example, a typical data set of an $O_3$ experiment is displayed in Figure 2 (experiment #3 in Table 1, no denuder, 5 % *RH*). The CCN-activity is presented as colored crosses (right bottom axis), the particle number concentration as a black line (left bottom axis), the $O_3$ concentration as a blue line (left top axis), and *RH* as a red line (right top axis), all as a function of experimental duration (bottom axis).

While the experiment was conducted for about 18 h in total, individual time frames can be distinguished and are indicated by the capital letters A to D – the four modes of operation - in Figure 2. Before the start of each experiment it was ensured that the aerosol chamber was particle-free, i.e. a particle concentration of $< 1$ cm⁻³ at the CPC downstream of the aerosol chamber. Then the aerosol chamber was filled with the size-selected particles at a flow rate of 23 L·min⁻¹. During the first hour, a subset of the aerosol stream (approx. 4 L·min⁻¹) bypassed the aerosol chamber for the determination of the baseline characteristics of the aerosol. This period is indicated by the letter A (bypass). Data acquisition from the bypass was completed after completion of one full SS scan in the CCNC. From then on, the sample was extracted from the aerosol chamber, while the filling of the aerosol chamber continued. This period is indicated by the letter B (filling). Due to running the aerosol chamber in CSTR mode, a dynamic equilibrium was established within the aerosol chamber after a certain time. Once sufficient data of the aerosol in that stage was acquired (at least 3 full scans with the CCNC) another bypass sampling period was started. This period is indicated by the letter C (bypass) in Figure 2. Data of this period were used to ensure that no changes in the particle production caused changes in the particle properties since the start of the experiment. Similar to the procedure in period A, one full SS scan in the CCNC was executed before this sampling period was completed by returning to sampling from behind the aerosol chamber. Simultaneous to changing the sampling extraction location, the supply of fresh soot particles into the aerosol chamber was stopped. The particle-containing inlet flow was replaced by particle- and VOC-free compressed air. This period is indicated by the letter D (flushing).

The experimental procedure is reflected by the change in particle number concentration (black line, left bottom axis) presented in Figure 2. The values present the concentration within the sampling line just in front of the measurement instruments.

Therefore, values recorded within the bypass periods A and C present the particle number concentration in the bypass section and values recorded within the periods B and C present the particle concentration in the reaction chamber. At the beginning of period B, an increase in particle number concentration is recorded asymptotically approaching a plateau in the dynamic equilibrium. The particle loss rate within the aerosol chamber is significantly higher compared to the bypass section, therefore, the particle number concentration within the plateau is lower compared to the periods A and C. Throughout the flushing period D, an exponential decay of the particle number concentration is recorded in accordance with theoretical expectations as no fresh particles are supplied to the aerosol chamber. The experimental conditions within this period can be considered similar to standard batch experiments. Monitoring and active control of the particle number concentration, *RH,* and temperature within the feed-in flow as well as $O_3$ concentration, RH, and pressure within the aerosol chamber ensured consistent experimental conditions within the aerosol chamber. The $O_3$ concentration (blue line) as well as *RH* (red line) within the aerosol chamber are shown in the top panel in Figure 2, illustrating the constant conditions throughout the duration of the entire experiment. The experimental conditions investigated within this study span a multidimensional space as $O_3$ (0 or 200 ppb), *RH* (5 or 75 %), and gas-phase VOC content (sample denuded or undenuded) were varied. Each of these settings was repeated at least twice. A summary of the experimental conditions is shown in Table 1.

The experiments conducted in summer 2016 were executed at room temperature, which varied from day to day. During the experiments in summer 2018, the temperature in the aerosol chamber was actively controlled and held at 25 °C in addition to the room temperature being controlled and maintained at 23 °C. The chamber temperature ensured constant reaction conditions and the room temperature ensured constant operating conditions of the measurement instruments.

**Table 1: Summary of all experimental results. The type of soot sample is given in the first column. The experimental conditions, the experiment number, and the experiment date are given in the second, third, and fourth columns, respectively. Experimental activation times at the respective SS are given in the following 9 columns. The reaction temperature and the $O_3$ concentration are given in the last two columns. The values in brackets denote the standard deviation (SD).**

| soot type | description | # | date | $t_{act}$/min; at different $SS_{act}$ 0.20 % | 0.30 % | 0.40 % | 0.60 % | 0.80 % | 1.00 % | 1.20 % | 1.40 % | 1.60% | T/°C (SD) | $O_3$/ppb |
|---|---|---|---|---|---|---|---|---|---|---|---|---|---|---|
| CAST-Brown (CBW) | $O_3$ + 5% $RH$ + charcoal denuder | 1 | 11. Aug 16 | - | - | 577 | 552 | 550 | 524 | 451 | 389 | * | 24.14 (0.04) | 182 |
| | | 2 | 12. Aug 16 | - | - | - | 523 | 525 | 522 | 512 | 384 | * | 23.87 (0.04) | 192 |
| | $O_3$ + 5% $RH$ | 3 | 05. Aug 16 | 617 | 524 | 491 | 420 | 334 | 248 | 189 | 152 | * | 28.10 (0.10) | 205 |
| | | 4 | 06. Aug 16 | - | 516 | 504 | 445 | 388 | 295 | 228 | 185 | * | 27.06 (0.11) | 208 |
| | | 5 | 21. Aug 16 | - | - | - | - | - | 498 | 362 | 267 | * | 24.62 (0.04) | 184 |
| | $O_3$ + 75% $RH$ charcoal denuder | 6 | 09. Aug 16 | - | - | - | - | - | 552 | 424 | 296 | * | 24.79 (0.02) | 214 |
| | | 7 | 10. Aug 16 | - | - | - | - | - | 557 | 539 | 442 | * | 24.50 (0.03) | 208 |
| | | 8 | 20. Aug 16 | - | - | - | - | - | | 537 | 372 | * | 24.74 (0.03) | 168 |
| | $O_3$ + 75% $RH$ | 9 | 03. Aug 16 | - | - | - | - | - | 328 | 301 | * | * | 27.03 (0.12) | 212 |
| | | 10 | 04. Aug 16 | - | - | - | 628 | 487 | 368 | * | * | * | 27.92 (0.11) | 208 |
| | | 11 | 25. Aug 16 | - | - | - | - | - | 498 | 371 | 312 | * | 23.14 (0.22) | 228 |
| | 5% $RH$ + charcoal denuder | 12 | 15. Aug 16 | | | | | | | | | * | 24.21 (0.03) | 0 |
| | | 13 | 16. Aug 16 | | | | | | | | | * | 24.25 (0.03) | 0 |
| | 5% $RH$ | 14 | 13. Aug 16 | | | | No CCN-activity was observed | | | | | * | 23.93 (0.01) | 0 |
| | | 15 | 14. Aug 16 | | | | | | | | | * | 24.04 (0.01) | 0 |
| | 75% $RH$ + charcoal denuder | 16 | 17. Aug 16 | | | | | | | | | * | 24.52 (0.02) | 0 |
| | | 17 | 18. Aug 16 | | | | | | | | | * | 24.64 (0.03) | $O_3$ spike |
| | 75% $RH$ | 18 | 08. Aug 16 | | | | | | | | | * | 24.96 (0.03) | 0 |
| | | 19 | 19. Aug 16 | | | | | | | | | * | 24.65 (0.03) | $O_3$ spike |
| CAST-Black | $O_3$ + 5% $RH$ + charcoal denuder | 20 | 10. Jul 18 | - | - | - | - | - | - | - | 725 | 552 | 25.10 (0.06) | 200 |
| | | 21 | 11. Jul 18 | - | - | - | - | - | - | - | 742 | 584 | 25.07 (0.05) | 200 |

-     No CCN-activity was detected because $t_{act}$ was higher than the maximum aging time.
\*     The highest SSs could not be reached due to technical limitations in the CCNC.

# 3 Data Analysis

## 3.1 CSTR mode versus batch mode

As can be seen in Figure 2, the transformation of fresh soot particles at an atmospherically relevant $O_3$ concentration demands multiple hours of reaction time before CCN-activity of the particles can be detected (after ~2 h). In batch mode operation, a reaction volume is first filled with the sample aerosol as fast as possible to achieve high homogeneity of the sample. After the desired starting concentration is achieved, further addition of the sample aerosol is stopped and the aging is initiated e.g. by addition of the oxidant. This point in time is generally referred to as $t = 0$ in such experiments. Analysis of the sample takes place while the reaction volume is flushed with sample-free gas. The aerosol chamber available at ETH Zurich has a volume of ~3 $m^3$. Therefore, it was not possible to achieve sampling times of up to 12 h at the flow rates demanded by the suite of instruments deployed if the aerosol chamber was operated in batch mode. With the aim to perform aging experiments at atmospherically relevant oxidant concentrations and to allow for atmospherically representative aging durations, the aerosol chamber was operated in CSTR mode. As mentioned previously, this mode of operation is characterized by a continuous addition of fresh aerosol simultaneous to a continuous extraction of the sample while the reaction conditions (e.g. oxidant concentration) are kept constant in the reaction volume.

While the entire aerosol is uniformly aged in batch experiments, the aerosol within a CSTR setup consists of a homogeneous mixture of differently aged aerosol particles. The continuous extraction of particle sample taking place concurrently to the addition of fresh particles causes fresh and old particles of varying residence times to be present simultaneously. Supported by the active mixing of the fan, the extracted sample consists of a homogeneous mixture of the particles in the aerosol chamber. The distribution of the particles in terms of their residence time within the aerosol chamber is well defined as it solely depends on the characteristics (e.g. volume) and operating conditions (e.g. flow rates) of the aerosol chamber operated in CSTR mode. The variability in residence times is referred to as Residence Time Distribution (RTD) under ideal conditions and Particle Age Distribution (PAD) under real conditions as will be discussed in the following section. A more detailed description of the experimental approach used here can be found in Friebel and Mensah (2019).

## 3.2 Activated Fraction

The CCN-activityof the soot particles is presented as activated fraction ($AF$), which is calculated by dividing the number of activated soot particles detected by the CCNC by the total number of particles entering the CCNC. The total number of particles (black line in Figure 2) is calculated by integrating the size distribution data of the SMPS downstream of the aerosol chamber. Independent of the SS, $AF$ is 0 in both bypass periods (A and C), indicating no CCN-activityof the unaged soot particles. This finding is consistent throughout all of the experiments performed.

At the beginning of the filling phase (B), $AF$ is still 0 even though the sample is taken from the aerosol chamber volume. The measured $AF$ increase only after a certain time threshold and reaches a constant level similar to the evolution of the particle number concentration. The plateau phase indicates that the conditions within the aerosol chamber have reached steady state. The point in time that $AF$ deviates from 0 as well as the $AF$ value in the plateau phase are dependent on the SS. The time it takes for $AF$ to deviate from 0 is shorter and $AF$ reaches higher values with increasing SS.

When the experimental settings are switched to flushing (phase D), i.e. no fresh particles are supplied to the aerosol chamber any longer, a steep increase in $AF$ can be observed while the particle number concentration decreases exponentially. In theory, a maximum $AF$ of 1 should be reached at all SS levels if sufficient experimental aging time was permitted. In the case of the experiment shown in Figure 2, the experimental duration is sufficient to allow for an $AF$ of 1 at SS of 1.4 % and 1.2 % only.

Due to the CSTR concept, the evolution of *AF* over time is significantly different from experiments conducted in batch-mode aerosol chambers, therefore a different data analysis concept is required.

### 3.3 Activation time $t_{act}$

Generally, the critical supersaturation ($SS_{crit}$) is reported from batch experiments to present the CCN-activity of the particles. The $SS_{crit}$ is defined as the SS where an *AF* of 0.5 is reached at a specific time after the start of the experiment. This parameter cannot be extracted directly from CSTR data as presented herein. Instead, the new parameter the activation time ($t_{act}$) will be used as a reference parameter as has been introduced by Friebel and Mensah, (2019). Although $t_{act}$ and its corresponding activation supersaturation ($SS_{act}$) are not identical to the $SS_{crit}$, after a defined aging time, both data sets are comparable.

While the transformation caused by the $O_3$ oxidation can be considered a continuous process, the change in CCN-activity of an individual soot particle at a defined SS is discontinuous and can be referred to as a non-gradual transition or a transition within a binary system as a particle is either inactive or active. In this context, $t_{act}$ represents the minimum aging time a single soot particle requires to cross a certain transformation threshold. The $t_{act}$-concept is valid for any transformation process involving a threshold. In the specific case presented herein, this process corresponds to a change in CCN-activity. As can be seen in Table 1, $t_{act}$ is dependent on the SS. The higher SS, the shorter $t_{act}$. In other words, the higher SS, the less transformation and therefore the less time is needed to cause CCN activation of a particle. The *AF* can, therefore, be defined as the fraction of particles that is older than $t_{act}$. Assuming ideal conditions, $t_{act}$ in steady state can be calculated following eq. (1), with $\tau_{CSTR}$ being the hydrodynamic residence time which is defined as the ratio of the volume of the CSTR ($V_{CSTR}$) to the total flow rate through the volume ($\dot{V}$) (Friebel and Mensah, 2019).

$$t_{act} = -\ln(AF) \cdot \tau_{CSTR} \tag{1}$$

$$\tau_{CSTR} = \frac{V_{CSTR}}{\dot{V}} \tag{2}$$

### 3.4 Particle losses

Knowledge of the particle age distribution (PAD) inside the aerosol chamber is required for the extraction of $t_{act}$. In case particle losses are negligible, the PAD within the aerosol chamber is identical to the residence time distribution (RTD) of the particles as shown in the equation below (eq. (3)).

$$RTD(t) = e^{\frac{-t}{\tau_{CSTR}}} \tag{3}$$

If particle losses occur, the PAD deviates from the RTD. As apparent by the reduced particle number concentration within the aerosol chamber in steady state compared to the bypass measurements, significant particle losses occurred in the aerosol chamber. In fact, there were two processes occurring simultaneously which cause a reduction in particle number concentration. First, the particle loss to any surface within the aerosol chamber e.g. the aerosol chamber walls. Since this loss process can be described by a first-order loss kinetic, the loss rate ($k_{loss}$) is directly proportional to the particle number concentration. Second, the particle removal due to sample extraction ($k_{CSTR}$), which can be considered a loss process as well. Since both processes follow the same kinetic they can be combined by introducing the effective particle loss rate $k_{age}$ and its reciprocal, the particle lifetime $\tau_{age}$. To obtain $k_{age}$ for the two first-order particle loss processes the individual loss rate constants have to be summed up as shown in eq. (4) below.

$$k_{age} = k_{CSTR} + k_{loss} = \frac{1}{\tau_{CSTR}} + \frac{1}{\tau_{loss}} = \frac{1}{\tau_{age}} \tag{4}$$

$$PAD(t) = e^{\frac{-t}{\tau_{age}}} \tag{5}$$

Here, the particle wall loss rate constant is $k_{loss}$. The particle flush rate constant ($k_{CSTR}$) is the inverse of the hydrodynamic residence time $\tau_{CSTR}$. The particle age distribution (PAD) can finally be calculated by substituting $\tau_{CSTR}$ in eq. (1) by the real particle lifetime ($\tau_{age}$) from eq. (4) leading to eq. (5) as shown above. The individual loss rates were determined in every single experiment according to the following procedure. The decay in particle number concentration recorded during flushing (period D) was defined as the total loss rate $k_{age}$. Assuming ideality of the set-up, the experimental flush rate is expected to be equal to the theoretical flush rate ($k_{CSTR}$), which can be calculated based on the flow rates (eq. (2)). Therefore, the difference between the experimental value and the theoretical value corresponds to the wall loss rate $k_{loss}$.

During the first campaign (Summer 2016) when the majority of CBW experiments were performed, $\tau_{age}$ ranged from 96 to 102 min and $\tau_{loss}$ from 500 to 800 min. During the second campaign (Summer 2018) when the CBK experiments were performed, $\tau_{age}$ ranged from 100 to 108 min. The increased particle lifetime was a result of a reduced wall loss rate and therefore an increased $\tau_{loss}$ ranging from 1000 to 2000 min. We attribute this pronounced change in wall loss rate within the aerosol chamber to the fact that the aluminum fan was gold-plated prior to the second campaign.

## 4    Results

### 4.1    CCN-activity

Following the discussion in the previous section, the real $t_{act}$ can be calculated by replacing the ideal hydrodynamic residence time ($\tau_{CSTR}$) in eq. (1) with the real particle lifetime ($\tau_{age}$) from eq (4) leading to eq (6) shown below.

$$t_{act} = -\ln(AF) \cdot \tau_{age} \tag{6}$$

Table 1 provides an overview of all experiments performed, including the various experimental conditions employed. Significant CCN-activation was observed only in experiments with an $O_3$ concentration of ~200 ppb. Contrary to the impact of $O_3$, neither elevated humidity conditions nor VOC denuding had an effect detectable with the instrumentation deployed.

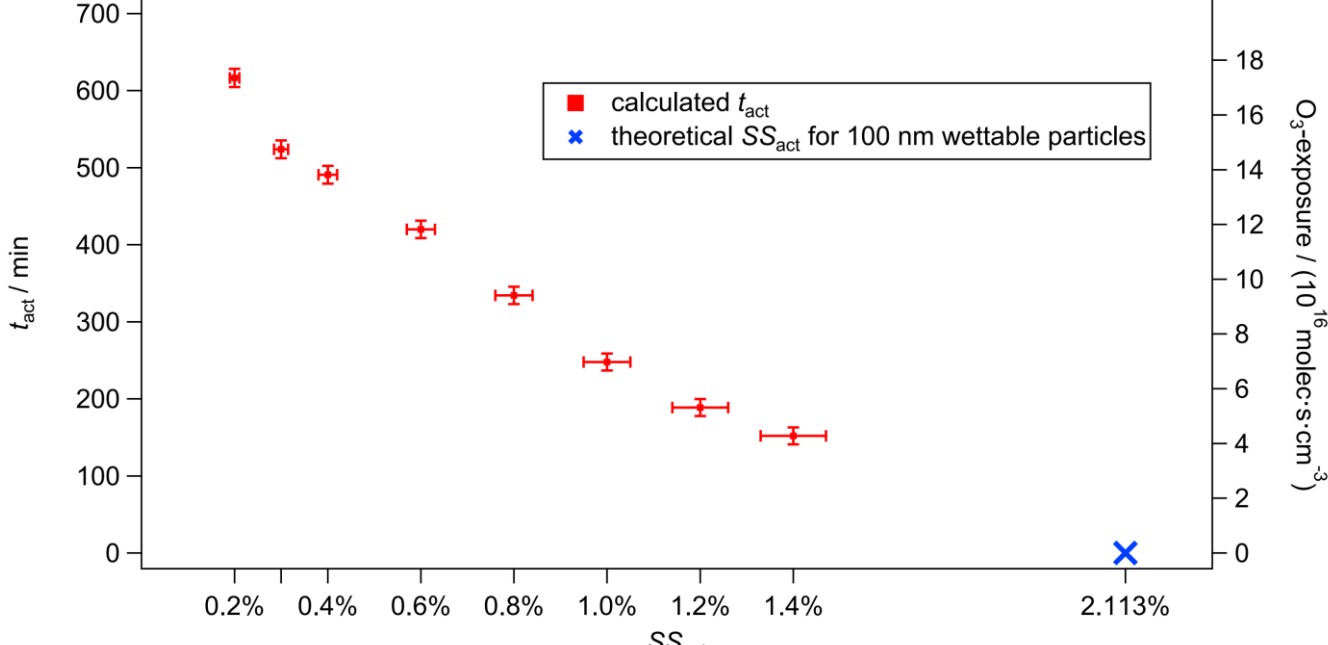

**Figure 3** $t_{act}$ **(left axis) and $O_3$-exposure (right axis) vs. theoretical (blue cross) and experimental (red markers) $SS_{act}$ for the experiment from 05.08.2016 with CBW and $O_3$ (200 ppb). The vertical bars represent the total error of $t_{act}$ of ± 12 min, which**

originates from the instrumental errors of the CPC and CCNC. The horizontal bars show the 5 % uncertainty in the SS of the CCNC (Rose et al., 2008).

In Figure 3, the activation time ($t_{act}$, left axis) as a function of the activation supersaturations ($SS_{act}$, bottom axis) is presented for the same experiment as shown in Figure 2 (100 nm CBW, 200 ppb $O_3$, $RH$ 5 %, no denuding; experiment #3 in Table 1). The right axis shows the cumulative $O_3$-exposure in molec·s·cm$^{-3}$, which is the product of the $O_3$ concentration and the exposure time. In our case, the exposure time corresponds to $t_{act}$. While an activation time of 152 min was determined at an $SS_{act}$ of 1.4 %, the activation time was more than three times higher at an atmospherically relevant $SS_{act}$ of 0.3 % (524 min). An activation time of 617 min (>10 h) was determined at an $SS_{act}$ of 0.2 % highlighting the capability of achieving atmospherically relevant aging durations within an aerosol chamber run in CSTR mode. The activation times are calculated from the $AF$ in steady state according to eq. (6). The vertical error bars represent an instrumental uncertainty of $\pm$ 12 min calculated by error propagation from the instrumental uncertainties of the CCNC and CPC. The horizontal error bars represent a 5 % uncertainty in the SS inside the CCNC following the recommendations of Rose et al. (2007). Details on the error calculation are presented in Friebel and Mensah, (2019).

As can be seen in Figure 2, the increase of $AF$ from zero appears later with decreasing SS. Similarly, lower $AF$s are determined for lower SS in steady state. Both aspects correspond to an increase in activation time with decreasing SS. In other words, the modification caused by $O_3$ oxidation needs a longer time to allow for CCN activation at lower SS s.

As mentioned previously, no CCN activation could be determined without exposure to $O_3$. Nevertheless, the $SS_{act}$ and $t_{act}$ for CBW particles without $O_3$ exposure can be estimated using kappa-Köhler theory (Petters and Kreidenweis, 2007). A spherical particle with a diameter of 100 nm that is non-hygroscopic (kappa = 0), but fully wettable (contact angle = 0°) should activate at a SS of 2.1 % (blue cross in Figure 3). As can be seen, this theoretical data point aligns well with the experimental results. Overall, an almost linear decrease in $t_{act}$ was determined with increasing $SS_{act}$ as can be seen in Figure 3. However, this is a theoretical approach as there is no scientific evidence that would support a linear correlation between oxidation with $O_3$ and $SS_{act}$. Furthermore, it is unclear if the soot used here is fully wettable or only partially wettable (contact angle > 0°) which would demand a higher $SS_{act}$ at $t_{act}$ = 0 min.

In all experiments with an $O_3$ concentration of 200 ppb, the same trend of decreasing $t_{act}$ with increasing $SS_{act}$ was observed independent of soot type, $RH$ and VOC conditions. Despite the similarity in the trend, the individual values of $t_{act}$ at the same $SS_{act}$ change by up to a factor of 2 between experiments of the same soot type and at the same experimental conditions in terms of $O_3$ concentration, $RH$ and VOC concentration. For example, looking at $t_{act}$ for CBW particles at 200 ppb $O_3$, 5 % RH, without denuding, and at an $SS_{act}$ of 1.4 % leads to a value of 152 min for experiment #3 and 267 min for experiment #5. These two values differ by a factor of 1.8. This deviation is significantly higher than the instrumental uncertainty of $\pm12$ min discussed above. Further analysis of the experimental conditions in summer 2016 and additional test experiments indicate that the average reaction temperature inside the aerosol chamber had a significant impact on the activation time. With increasing reaction temperature, shorter $t_{act}$'s were determined. Because attempts to control the room temperature by air conditioning in summer 2016 were not sufficient to keep the reaction temperature stable, the temperature of the aerosol chamber itself was actively controlled in the experiments performed in summer 2018.

## 4.2    CAST-Black

During the second measurement campaign in summer 2018 a second soot type (CAST black; CBK) was investigated. The particles of this soot type are characterized by a significantly reduced OC content compared to CBW particles as presented in Table 4. CBK particles were exposed to 200 ppb $O_3$ at $RH$ of 5 % with a charcoal denuder in line. As no impact of $RH$ and VOC could be determined in the campaign in summer 2016, these parameters were kept constant in all CBK experiments in summer 2018. Nevertheless, the experimental setup was improved by implementing a direct temperature control of the chamber (see section 2.5) and by reducing particle losses (see section 3.4).

A significant difference in CCN-activity upon $O_3$ exposure was determined between the two soot types - CBW and CBK. As can be seen in Table 1, CBK particles show much lower CCN-activity than CBW particles. CBK particles had to be oxidized for 725 to 742 min in order to show CCN-activity at an $SS_{act}$ of 1.4 %, which corresponds to an increase in $t_{act}$ by a factor of 2 to 4 times compared to CBW particles(experiment #20 and #21 vs #1 and #2 in Table 1). Considering similar minimum aging durations/$t_{act}$'s, CBW particles activate at an $SS_{act}$ of 0.4 % after 552 and 523 min (#1 and #2) while CBK particles demand an $SS_{act}$ of 1.6 % for activation after 552 and 584 min (#20 and #21), respectively. Overall, no CCN-activity of CBK particles could be detected at atmospherically relevant SS (0.3% - 0.8 %; Pruppacher and Klett, 2010) within the maximum aging time of up to 12 h..

### 4.3 $O_3$ Spike Experiments

For the experiments #17 and #19, soot particles were not exposed to $O_3$ while the aerosol chamber was filled. Only after switching to the flushing mode the $O_3$ concentration was ramped to 200 ppb within approximately 30 min. Once this concentration threshold was reached no further $O_3$ was added. The $O_3$ concentration decayed exponentially reaching a value of 50 ppb within 120 min after the $O_3$ supply to the aerosol chamber was switched off. Despite the temporary exposure to $O_3$, no CCN-activity at any SS could be detected within the remaining experimental duration of 6 hours. However, an increase in the particle mean diameter of 3 nm was detected while the $O_3$ was added to the chamber.

### 5 Discussion

In an attempt to attribute the change in CCN-activity to the heterogeneous oxidation with $O_3$ we investigated various parameters. These parameters include particle size, reaction temperature, relative humidity, and VOC content of the sample. The particle size was determined by DMA measurements. Size distribution measurements of the particles before and after aging in the aerosol chamber revealed no substantial restructuring such as compaction of the particles. To the contrary, a slight growth upon $O_3$ exposure was detected in the range of 3 nm. Such growth of particles has already been reported by Fendel et al. (1995) for metal and spark discharge graphite particles and by Kotzick et al., (1997) for spark discharge graphite particles. A detailed analysis of this aspect is beyond the scope of this paper.

Experiments performed during the measurement campaign in summer 2016 were executed at room temperature without active temperature control of the aerosol chamber. Despite an air conditioning unit being installed, the difference between the coldest and the warmest average daily temperature measured throughout the campaign was greater than 5 K. Referring to the results of two CBW experiments executed at the same conditions (200 ppb $O_3$, 5 % $RH$, without denuder; experiment #3 to #5 in Table 1) , it can be seen that a decrease in the average chamber temperature is associated with an increase in activation time. Such temperature dependency is in accordance with the expected impact of temperature on the reaction speed following the van't Hoff rule. It is known from model simulations and experimental studies that the $O_3$ oxidation of polycyclic aromatic hydrocarbons (PAH) and organic molecules with C=C double bonds require an activation energy of 40 to 80 kJ·mol$^{-1}$ (Berkemeier et al., 2016; Lee et al., 2009; Pöschl et al., 2001; Stephens et al., 1989). Even though many different compounds can be found on soot surfaces, PAH's are considered to be a good reference compound (Slowik et al., 2004). A temperature change by 5 K would change the reaction speed and therefore $t_{act}$ by a factor of 2. The deviations in $t_{act}$ determined experimentally are within the same order of magnitude as the theoretical calculations supporting the presumed impact of reaction temperature.

Investigation of the $RH$ conditions revealed neither a short-term nor a long-term effect within the range 5 - 75 %. Changes in the particle morphology could be considered as a short-term effect. Contrary to the impact of $O_3$, no significant change of the particle diameter could be detected upon exposure of the particles to elevated $RH$ conditions. Overall, our findings are

supported by Mahrt et al., (2018) who showed that the water uptake on CBW and CBK particles does not exceed the adsorption of one monolayer at *RH* below 90 %. Long-term exposure of the particles to elevated *RH* conditions showed no impact on the CCN-activity even after up to 12 h, which is independent of the soot type investigated within this study.

Homogeneous $O_3$ oxidation of VOCs can lead to semi-volatile reaction products which in turn can condense onto pre-existing particles and thereby modify the particle's CCN-activity (Wittbom et al., 2014). Because the VOC concentration within the aerosol chamber could not be determined directly, the impact of VOCs emitted by the miniCAST was evaluated by implementing a charcoal denuder into the experimental setup. No impact on CCN-activity or particle size could be determined for experiments with and without the denuder in line.

### 5.1    CCN-activity

In Figure 3 the $t_{act}$ as a function of $SS_{act}$ is presented. As can be seen, increasing $SS_{act}$ are associated with decreasing $t_{act}$'s . The uncertainty in the determination of $t_{act}$ is ± 12 min and originates from the instrumental errors of the CPC and CCNC as reported by Friebel and Mensah (2019). Therefore, relative uncertainties in $t_{act}$ and the calculated $O_3$-exposure are below 10 %. Compared to the uncertainties reported for the OH-exposure from different OFRs which are on the order of a factor of 5 (Lambe et al., 2011; Simonen et al., 2017), the uncertainties for the approach used here are significantly smaller.

While the distinct mechanism that leads to the significant change in CCN-activity of oxidized soot (e.g. inverse Kelvin effect, formation of soluble or surface-active compounds) cannot be identified, it can be ruled out that the change is due to a growth of the particle diameter. The average diameter increase (CBW: + 3 nm; CBK: + 1.5 nm) is too small to have a decisive impact on the CCN-activity. Furthermore, the growth of the diameter occurs on a time scale of max. 30 min and is therefore much faster than the change in CCN-activity which occurs over a time scale of multiple hours.

Overall, the soot particles show more pronounced CCN-activation after exposure to $O_3$ than has been reported previously in the literature. It should be mentioned that this assertion is qualitative, because the particle sizes and particle compositions vary across the different studies. Nevertheless, the cumulative $O_3$-exposure, the product of the $O_3$ concentration and the exposure time, can be taken as a metric for comparison. On that basis, 100 nm diameter CBK particles in our study ($SS_{act}$ = 1.6 % after $4.9 \times 10^{16}$ molec·s·cm$^{-3}$ $O_3$-exposure) show CCN-activity within the same order of magnitude as 150 nm kerosene diffusion flame soot particles investigated by Grimonprez et al., (2018) ($SS_{crit}$ = 1.4 % after $10 \times 10^{16}$ molec·s·cm$^{-3}$ $O_3$-exposure) and as 222 nm ethylene premix flame soot particle ($SS_{crit}$ = 1.5 % after $5 \times 10^{16}$ molec·s·cm$^{-3}$ $O_3$-exposure; Lambe et al., 2015).

The differences could be attributed to the different chemical compositions of the soot particles as well as the different experimental setups but indisputable statements cannot be made on the basis of the data currently available. Experiments by Lambe et al., (2015) were performed at an $O_3$ concentration of up to 20 ppm and exposure times of 100 s. In contrast to that, the approach presented herein allows for atmospherically relevant oxidant concentrations (200 ppb) and exposure times (up to 12 h). Note the comparison approach in terms of the cumulative $O_3$-exposure performed here is valid only if the reaction speed is directly proportional to the $O_3$ concentration i.e. follows a first-order reaction kinetic. The validity of this assumption cannot be verified on the basis of the data presented herein.

The differing activation times of CBW and CBK particles investigated in the same experimental setup indicate an impact of the chemical composition. $O_3$-exposures higher by a factor of 2 to 4 are required to cause the same level of activation for CBK particles compared to CBW particles of the same size and experimental conditions. In view of the abundance of soot particles in the atmosphere, the increase in CCN-activity of CBW and CBK particles due to heterogeneous oxidation of soot particles can be considered as atmospherically relevant. A linear extrapolation to atmospheric $O_3$ background concentration levels of 20 to 45 ppb (Hough and Derwent, 1990; Vingarzan, 2004) shows that CBW and CBK particles would become CCN-active at 0.3 % SS after 2 to 4 days and 4 to 16 days, respectively. These values lie within the range of the average atmospheric lifetime of one week (Textor et al., 2006) and indicate that this aging pathway could be a significant source of CCN-active soot particles within the atmosphere.

## 6    Atmospheric relevance

Similar as in a CSTR aerosol chamber, particles are constantly emitted into the atmosphere as well as constantly removed from the atmosphere except in case of plume events. As a result, a mixture of particles at different aging stages is present in the atmosphere. From this perspective, the atmosphere can be approximated to be a CSTR in steady state. This approximation indicates that CSTR data is at least as suitable for parameterizations in global climate models data obtained from other experimental setups. We performed three experiments with the global aerosol-climate model ECHAM6.3-HAM2.3 (Neubauer et al., 2019; Tegen et al., 2019) to evaluate if the change in CCN-activity of soot particles due to heterogeneous $O_3$ oxidation has an impact on the cloud droplet number concentration (CDNC) and therefore on cloud properties from a global perspective. The size distribution of atmospheric aerosol particles in ECHAM6.3-HAM2.3 is described by seven log-normal modes (four internally mixed modes and three externally mixed modes), the particle number concentration, and the mass mixing ratio of up to five aerosol components (sulfate, BC, particulate organic matter (POM), sea salt, mineral dust). While the structure of the size distribution is prescribed, the particle number concentration and the mixing ratio of each component are computed prognostically for each mode (for details see Tegen et al., (2019)). All BC emissions (fossil fuel, bio-fuel, biomass burning) and the POM emissions from fossil fuel are emitted into the externally mixed Aitken mode. Sixty-five percent of POM emissions from bio-fuel, biomass burning and biogenic secondary organic aerosols are emitted into the internally mixed modes and 35 % of these emissions are considered insoluble and emitted into the externally mixed Aitken mode (Zhang et al., 2012). In the standard setting of ECHAM6.3-HAM2.3, which was used for the reference (REF) experiment, only aerosol particles in the internally mixed modes can serve as CCN. Their activation is further dependent on their size and hygroscopicity. The activation of aerosol particles to cloud droplets occurs following a Köhler-theory based parameterization of Abdul-Razzak and Ghan, (2000) (for details see Stier, 2016). Our REF-experiment is almost identical to the E63H23-10CC-experiment presented in Neubauer et al., (2019) with the exception of three differences. First, while the modeling experiment simulates a period of 20 years, the aerosol emissions are based on the year 2008 for all years within our simulation. Second, 31 hybrid-sigma vertical levels were used in our simulations vs. 47 in Neubauer et al. (2019) as our focus is on the troposphere. Sigma-hybrid means that the levels close to the surface follow the topography (sigma) while the levels at higher altitude follow the pressure evolution.). Third, the activation parameterization was updated as in the standard version the calculation of the average solubility of the individual modes yielded artificially low values (Neubauer et al., 2019). Two sensitivity modeling experiments were performed. which were identical to the REF experiment except that In these sensitivity experiments the REF experiment was extended by allowing BC and POM particles in the externally mixed Aitken mode to activate to cloud droplets using a parameterization developed based on the results from the CSTR aging experiments. Details of this parameterization will be described briefly in the next section and further details can be found in the appendix (section 8.2.).

### 6.1    Parameterization of experimental results

Following the scheme of Abdul-Razzak et al., (1998), the parameter B in eq. (10) and (12) therein, entails the solubility of the aerosol particles in the model and is calculated from the van't Hoff factor $\nu$ and the osmotic coefficient $\varphi$. These parameters ($\nu$ and $\varphi$) are not directly available from our measurements, but their product can be calculated taking the particle diameter ($d$ = 100 nm) and the activation supersaturation ($SS_{act}$ = 0.3 %) of the particles of interest (BC and POM particles in the externally mixed Aitken mode) into account as shown in eqs (10-12) in appendix 8.2. While the product of $\nu$ and $\varphi$ is kept constant at all time steps and grid boxes for, the fraction of CCN-active BC and POM particles in the externally mixed Aitken mode per grid box ($X_{CCN}$) is calculated for each grid box and time step individually. According to the definition of $t_{act}$, $X_{CCN}$ is equal to the fraction of these specific particles that are older than $t_{act}$. In both sensitivity experiments, individual reference activation times ($t_{act,ref}$) representative of CBW and CBK particles were chosen for BC and POM particles in the externally mixed Aitken mode. $t_{act,ref}$ is defined as the minimum aging time after which the particles show CCN-activity at 0.3 % SS and an $O_3$ concentration of 200 ppb. In the first case $t_{act,ref}$ is equal to 10 h, which is derived from the experimentally determined $t_{act}$ for

CBW. In the second case $t_{act,ref}$ is equal to 50 h, which is set based on an extrapolation from experimentally determined $t_{act}$ values for CBK. The effective $t_{act}$ is calculated from $t_{act,ref}$ and the $O_3$ concentration in the grid box at each time step assuming a first-order reaction kinetics with respect to $O_3$ in accordance with Friedman et al., (2011), Lambe et al., (2015) and Grimonprez et al., (2018) as shown below in eq. (7). More information can be found in the appendix 8.2.

$$t_{act} = t_{act,ref} \cdot \frac{200 \text{ ppb}}{[O_3]} \tag{7}$$

5 Adapting eq. (3), which describes the residence time distribution in a CSTR, allows for the estimation of the particle age distribution in the atmosphere as presented in eq. (8). The ideal mean particle lifetime ($\tau_{CSTR}$) is replaced by the average atmospheric life time ($\tau_{atm}$) of soot particles, i.e. 7 days according to Textor et al., (2006).

$$PAD_{atm}(t) = e^{\frac{-t}{\tau_{atm}}} \tag{8}$$

Integration of the atmospheric particle age distribution $PAD_{atm}(t)$ from $t = t_{act}$ to $t = $ infinity (eq. (9)) yields $X_{CCN}$, the fraction of CCN-active BC and POM particles in the externally mixed Aitken mode.

$$X_{CCN} = \int_{t=t_{act}}^{t=\infty} PAD_{atm}(t) \ dt = e^{\frac{-t_{act}}{\tau_{atm}}} \tag{9}$$

10 Please note that the increase of the soot particle's hygroscopicity due to oxidation with $O_3$ increases the soot particle removal rate from the atmosphere, e.g. due to a higher wet-deposition rate. As a result, the average atmospheric lifetime of soot particles decreases. However, the reduction of the soot particle lifetime was below 2% in both scenarios. Since this lifetime reduction is statistically not significant, its impact on the CCN burden and CDNC was not considered within this study.

## 6.2   Results – modeling

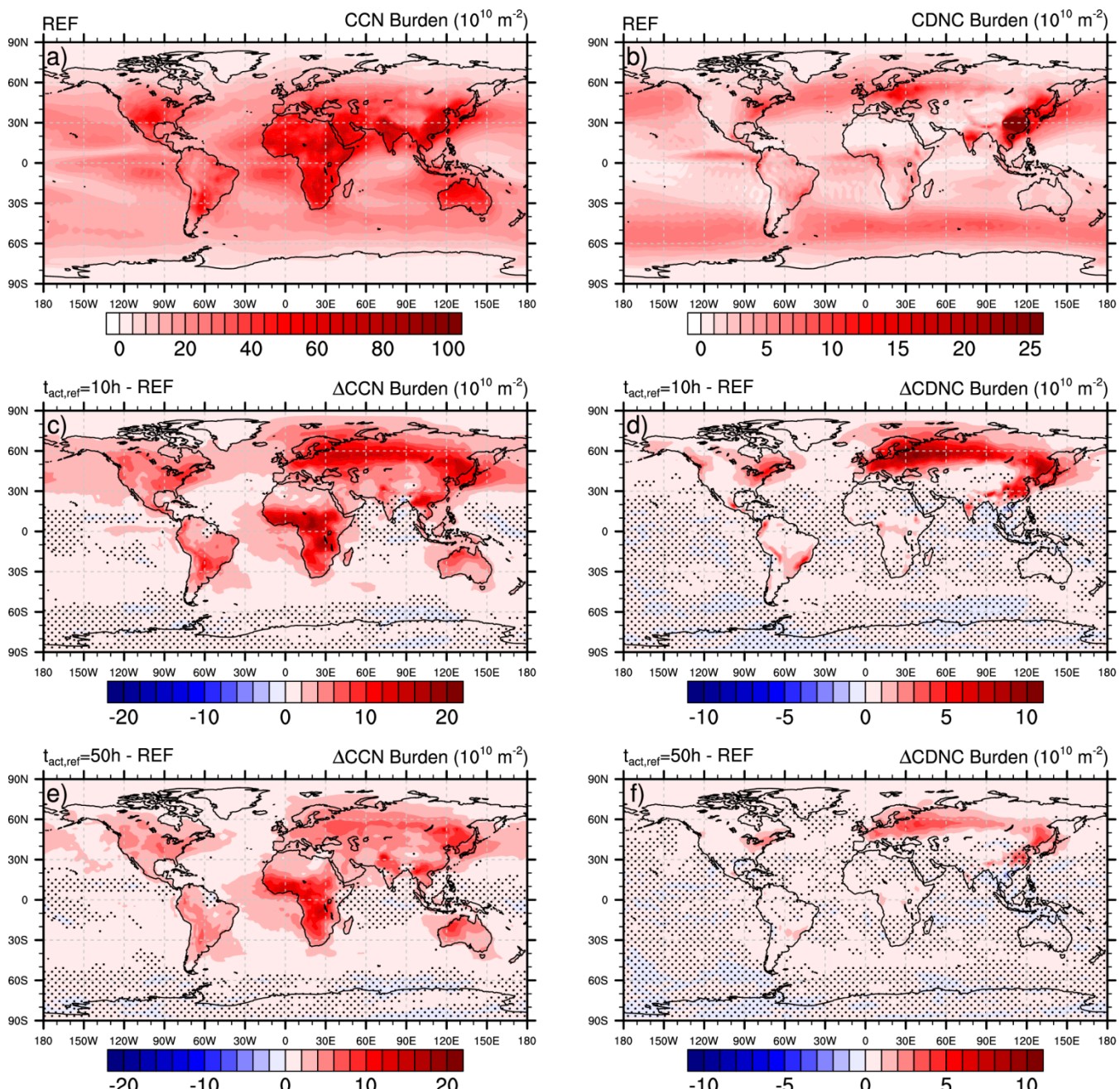

**Figure 4 20-year mean vertically integrated burden of cloud condensation nuclei (CCN; left column figures a, c, e) and cloud droplet number concentration (CDNC; right column; figures b, d, f), respectively,) of the reference simulation (REF; top row; figures a, b) and the change due to heterogeneous $O_3$ oxidation of soot particles for two reference activation times $t_{act,ref}$ = 10 h (middle row; c, d) and $t_{act,ref}$ = 50 h (bottom row; e, f). Blue colors indicate a reduction and red colors indicate an increase in the respective parameter. Note, for better perceptibility of the important features, hatching indicates statistically not significant differences (< 95 %).**

The results of the REF experiment and the sensitivity experiments are presented in Figure 4. While the top row presents the 20-year mean vertically integrated cloud condensation nuclei (CCN burden; left, a) and cloud droplet number concentration (CDNC burden; right, b) in the REF experiment, the two other rows present the change in these parameters taking the effect of the heterogeneous $O_3$ oxidation of soot particles into account. The middle row (Figure 4 c, d) presents changes in the CCN burden and CDNC burden assuming a reference activation time of 10 h, which can be considered representative for the activation behavior of brown carbon particles. Changes due to the contribution of black carbon particles are presented in the bottom row (Figure 4 e, f) with $t_{act,ref}$ = 50 h as a representative value. To allow for a better perceptibility of the important features within the figures, hatching indicates statistically not significant differences at the 95 % significance level. A two-

tailed unpaired Student's $t$-test is computed for each model grid box from annual mean values (20 for each experiment). The false discovery rate of the statistical significance is controlled following Wilks, (2016).

**Table 2: CCN mean burden of the three simulations globally averaged and averaged over different regions on the Northern Hemisphere as well as the change due to consideration of CCN-active BC and POM particles in the externally mixed Aitken mode after heterogeneous $O_3$ oxidation.**

| Region | CCN burden/($10^{10}$ m$^{-2}$) | | | $\Delta$CCN burden/% | |
|---|---|---|---|---|---|
| | REF | $t_{act,ref}$ = 10 h | $t_{act,ref}$ = 50 h | $t_{act,ref}$ = 10 h | $t_{act,ref}$ = 50 h |
| global | 22.4 | 25.3 | 24.6 | 12.7 | 9.7 |
| >= 60 °N | 5.3 | 9.1 | 7.2 | 71.5 | 36.0 |
| >= 50 °N | 9.4 | 14.3 | 12.0 | 52.4 | 27.9 |
| >= 40 °N | 15.5 | 20.6 | 18.6 | 33.1 | 19.9 |

Figure 4 panels c) and e) present the change in CCN burden due to consideration of BC and POM particles in the externally mixed Aitken mode after heterogeneous $O_3$ oxidation. Independent of $t_{act,ref}$, the strongest increase in CCN burden can be seen in the Northern Latitudes as well as in the tropics, namely over the sub-Saharan African continent. Overall the increase is more pronounced in the case of $t_{act,ref}$ = 10 h, which is representative of the transition behavior of brown carbon as determined within our experiments with CBW particles. Taking CBW particles into account, the global mean CCN burden increases from 22.4 × $10^{10}$ m$^{-2}$ in REF to 25.3 × $10^{10}$ m$^{-2}$ (Table 2). This increase of 12.7 % is statistically significant. At this $t_{act,ref}$, the maximal regional increase can be determined in the latitudes north of 60 ° with a relative increase of more than 70 %. Taking the transition behavior of black carbon particles into account ($t_{act,ref}$ = 50 h), which has been determined by the investigation of CBK particles, the global mean CCN burden is determined to be 24.6 × $10^{10}$ m$^{-2}$. This increase of 9.7 % is statistically significant, but 3 percentage points less than in the case of $t_{act,ref}$ = 10 h. Investigating the regional impact, the relative increase still maximizes in the latitudes north of 60 ° but is about half as strong as in the case of $t_{act,ref}$ = 10 h. Regional changes in CCN burden are pronounced where either the emissions and atmospheric burden of BC and POM are large (e.g. tropics) or where CCN concentrations are relatively low (e.g. central to northern Europe and Asia). We hypothesize that in regions where many CCN are available in the internally mixed modes, the additional CCN in the externally mixed Aitken mode compete with the CCN from the internally mixed modes for the available water vapor. This competition is also considered in the parameterization of Abdul-Razzak and Ghan, (2000). As a result of this competition the annual mean values of CCN show the largest differences to the REF experiment in regions where the emissions of BC and POM are large (not shown) and CCN concentrations are relatively low in the REF simulation (see Figure 4 and S1 in the supplement).

**Table 3: CDNC mean burden of the three simulations globally averaged and averaged over different regions on the Northern Hemisphere as well as the change due to consideration of CCN-active BC and POM particles in the externally mixed Aitken mode after heterogeneous $O_3$ oxidation.**

| Region | CDNC burden/($10^{10}$ m$^{-2}$) | | | $\Delta$CDNC burden / % | |
|---|---|---|---|---|---|
| | REF | $t_{act,ref}$ = 10 h | $t_{act,ref}$ = 50 h | $t_{act,ref}$ = 10 h | $t_{act,ref}$ = 50 h |
| global | 3.2 | 3.8 | 3.5 | 17.8 | 8.9 |
| >= 60 °N | 1.5 | 3.0 | 2.0 | 93.0 | 30.3 |
| >= 50 °N | 2.8 | 4.8 | 3.6 | 73.3 | 27.1 |
| >= 40 °N | 3.7 | 5.6 | 4.4 | 52.8 | 21.0 |

Similar to the changes in CCN burden, the strongest increases in CDNC can be found over land on the Northern Hemisphere. However, over the tropics the increase in CDNC is much lower than the increase in the CCN burden. The difference is caused by the higher abundance of stratiform liquid clouds in mid latitudes compared to the tropics, which is indicated by the higher CDNC burden in mid latitudes (Figure 4b). Please note that ECHAM6.3-HAM2.3 accounts for CDNC from detrained cloud water of convective clouds but otherwise the convective cloud parameterization does not consider CDNC (Neubauer et al.,

2019). Therefore, our simulations could underestimate the impact of heterogeneous ozone oxidation of soot particles on CDNC, in particular where convective clouds are common like the tropics. The largest changes in CDNC occur below about 700 hPa, i.e. for low-level clouds (Figure S1). Again, the impact at $t_{act,ref}$ = 10 h is much more pronounced than at $t_{act,ref}$ = 50 h with a global mean CDNC burden of $3.8 \times 10^{10}$ m$^{-2}$ (+ 17.8 % compared to REF) and $3.5 \times 10^{10}$ m$^{-2}$ (+ 8.9 % compared to

5 REF), respectively (Table 3). The largest increases in liquid cloud droplets occur around 60 °N over Europe, Asia and North America causing almost a doubling (+ 93.0 %) in the case of $t_{act,ref}$ = 10 h and an increase by more than 30 % in the case of $t_{act,ref}$ = 50 h.

## 7    Conclusion

We successfully applied the CSTR approach for the investigation of the change in CCN-activity of two soot types. Here we
present the results of experiments in which soot particles were exposed to 200 ppb $O_3$ and varying levels of humidity for up to 12 h. The CSTR approach allows for a low particle input concentration (1000 to 1500 cm$^{-3}$) and size-selection of particles (e.g. at 100 nm).

We show that the heterogeneous $O_3$ oxidation is a process that can make soot particles CCN-active at atmospherically relevant SSs of 0.3 to 0.8 %. The general finding agrees with literature results underlining the applicability of the CSTR approach.
Nevertheless, the $SS_{act}$ in our experiments is significantly lower at the same $O_3$-exposures compared to results obtained in other experimental setups (Grimonprez et al., 2018; Lambe et al., 2015). The soot rich in OC (CBW) demanded 2 to 4 times less aging time ($t_{act}$ = 3 – 6 h at 1.4 % $SS_{act}$) than soot low in OC (CBK, $t_{act}$ = 12 h at 1.4 % $SS_{act}$). In contrast to $O_3$, no effect of $RH$ (up to 75 %) or denuding of the gas phase was observed. Instead, it was found that temperature fluctuations of 5 K inside the aerosol chamber have a strong impact on the activation time $t_{act}$ and were the largest single contributor to the experimental
uncertainties.

A test with a global aerosol-climate model, where a first-order kinetic was assumed, showed that the change in CCN-activity of soot particles that are not taken into account in the standard configuration can lead to statistically significant increases in CCN burden and CDNC burden. The strongest increases were observed where the soot burden was large, and/or the initial CCN concentration was rather low for both reference activation times investigated. In the case of the CDNC burden it is
additionally beneficial if CCN do not have to compete for water vapor and stratiform liquid clouds are frequent.

Both, the discrepancy in activation levels between studies using different experimental approaches and the initial $O_3$ adsorption detected by a particle diameter increase within minutes suggest that the underlying reaction mechanism might not be sufficiently well described by assuming first-order kinetics. Therefore, we suggest performance of tailored experiments with a focus on the effect of different $O_3$ concentrations as well as different temperatures. This might allow for further insight
into the reaction kinetics and improvement of the accuracy of extrapolations to atmospheric conditions.

Data availability:

The data presented in this publication can be downloaded from http://dx.doi.org/10.5281/zenodo.2541937 . The scripts to produce Figures 4 and S1 can be downloaded from http://dx.doi.org/10.5281/zenodo.3452036 .

Author contributions:

FF and AAM prepared the manuscript with contributions from PL, DN, UL, SDD and EM. FF, AAM, PL, SDD and EM designed and conducted the experiments. FF analyzed the experimental data. DN and UL designed and analyzed the model simulations. The Figures 1, 2, and 3 were produced by FF and the Figures 4 and S1 were produced by DN.

Competing interests:

The authors declare no competing interests.

Acknowledgements:

This research was funded by the Swiss National Science Foundation, SNSF-grant #PZ00P2_161343. Prem Lobo was supported by the Swiss National Science Foundation International Short Research Visits Grant (Project number: #IZK0Z2_168324). The ECHAM-HAMMOZ model was developed by a consortium composed of ETH Zurich, Max Planck Institut für Meteorologie, Forschungszentrum Jülich, University of Oxford, the Finnish Meteorological Institute and the Leibniz Institute for Tropospheric Research, and managed by the Center for Climate Systems Modeling (C2SM) at ETH Zurich. The computing time for this work was supported by a grant from the Swiss National Supercomputing Centre (CSCS) under project ID s652. We thank Zamin A. Kanji, Oliver F. Bischof, and Thomas Peter for their valuable discussions, and Ulrike Lohmann's group for their support.

## 8    Appendix

### 8.1    Experimental Setup.

**Table 4 miniCAST 4200 burner set points used to generate soot aerosol.**

| Soot type | Propane fuel (L·min⁻¹) | N₂ mixing (L·min⁻¹) | Oxidation Air (L·min⁻¹) | N₂ quench (L·min⁻¹) | Dilution Air (L·min⁻¹) | Fuel air ratio | C:O ratio |
|---|---|---|---|---|---|---|---|
| CBW | 0.06 | 0.25 | 1.42 | 7.5 | 20 | 1.03 | 0.31 |
| CBK | 0.06 | 0.00 | 1.55 | 7.5 | 20 | 0.95 | 0.28 |

### 8.2    CCN-activity of soot in ECHAM6.3-HAM2.3

Within ECHAM6.3-HAM2.3 seven log-normal modes are defined for the aerosol particle size distribution. The CCN-activity of each mode is characterized by component specific parameters as well as the particle size distribution within each mode according to Abdul-Razzak and Ghan, (2000). However, the model does not contain "soot particle" as a category but contains BC and POM as separate categories. The two categories BC and POM together represent the properties of soot with a lesser or higher amount of organic material, respectively. Therefore, they are modified together to represent the change in CCN-activity of soot particles due to heterogeneous $O_3$ oxidation.

So far BC and POM particles are considered to be not CCN-active within ECHAM6.3-HAM2.3, unless they are internally mixed with soluble components such as sulfate. Particles are transferred from the insoluble to the soluble mode when sufficient sulfuric acid gas condenses on insoluble particles to form a monolayer of coating, or by coagulation with soluble particles. The product of the van't Hoff factor $\nu$ and the osmotic coefficient $\varphi$ describes the solubility of the particles in water. Within ECHAM6.3-HAM2.3, $\nu \cdot \varphi$ is set to 0, because BC and POM particles are considered to be not CCN-active. However, based on the experimental results we presented, it is possible to calculate a new value of $\nu \cdot \varphi$ for BC and POM particles, which is then used in the model experiments.

With the eqs. (10), (11) and (12) taken from Abdul-Razzak et al. (1998) the product $\nu \cdot \varphi$ can be calculated from a defined particle diameter and its $SS_{act}$. The molar mass ($M_{BC/POM}$) as well as the density ($\rho_{BC/POM}$) of BC and POM are the same as in our model experiments and are given along with other constants in Table 5. We choose a $SS_{act}$ of 0.3 % and a diameter of 100 nm diameter as reference values. The corresponding $t_{act,ref}$ values are used in the computation of $X_{CCN}$ in the model. To calculate $\nu \cdot \varphi$, the fraction of CCN-active particles ($\epsilon$) is set to 1. The values for $\nu \cdot \varphi$ are then 0.050 and 0.753 for BC and

POM, respectively. Note, within the actual model experiments $\epsilon$ is not set to 1 but it represents the mass fraction of BC/POM in the externally mixed Aitken mode in order to calculate the average SS in the externally mixed Aitken mode.

Please note that the treatment of secondary organic aerosol (SOA) is simplified in ECHAM6.3-HAM2.3. During emission the soluble and insoluble fractions of SOA are assumed to condense immediately on the soluble Aitken and accumulation and insoluble Aitken mode respectively. However, He et al. (2016) accounted in their experiments for BC aging by condensation of SOA and found that their chemical aging mechanism still accounted for more than 50% of the BC aging rate at high latitudes (polewards of 60°N/S) and above 900 hPa.

$$SS_{act} = \frac{2}{\sqrt{B}} \cdot \left(\frac{A}{3 \cdot d}\right)^{3/2} \tag{10}$$

$$A = \frac{4 \cdot \sigma_{a/w} \cdot M_w}{R \cdot T \cdot \rho_w} \tag{11}$$

$$B = \frac{v \cdot \varphi \cdot \epsilon \cdot M_w \cdot \rho_{BC/POM}}{M_{BC/POM} \cdot \rho_w} \tag{12}$$

**Table 5 Parameters and constants used for the calculation of the CCN-activity of POM and BC-particles .**

| | | |
|---|---|---|
| $\sigma_{a/w}$ | 0.072 J·m$^{-2}$ | surface tension of water at 25 °C |
| $M_w$ | 0.018 kg·mol$^{-1}$ | molar mass of water |
| $\rho_w$ | 1000 kg·m$^{-3}$ | density of water |
| R | 8.314 J$^{-1}$·mol·K$^{-1}$ | universal gas constant |
| $T$ | 298.15 K | temperature |
| $d$ | 100 nm | reference particle diameter |
| SS | 0.3 % | reference SS |
| $\epsilon$ | 1 | mass fraction of soluble material |
| black carbon (BC) | | |
| $M_{BC}$ | 0.012 kg·mol$^{-1}$ | molar mass of black carbon |
| $\rho_{BC}$ | 2000 kg·m$^{-3}$ | density of black carbon |
| particulate organic matter (POM) | | |
| $M_{POM}$ | 0.180 kg·mol$^{-1}$ | molar mass of particulate organic matter |
| $\rho_{POM}$ | 2000 kg·m$^{-3}$ | density of particulate organic matter |

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
