# Peer review of "Impact of Isolated Atmospheric Aging processes on the Cloud Condensation Nuclei-activation of Soot Particles"

_Atmospheric Chemistry and Physics, 2019_

## Referee Comment (RC1) · Anonymous Referee #1 · 26 Aug 2019

**General comments**

This paper discusses the aging of soot particles by ozone oxidation in an environmental chamber. Aging times of up to 12 hours were achieved, and the data analysis follows recently published work by the same author to interpret results from the continuous-flow stirred tank reactor chamber. Speaking of that prior work (Friebel and Mensah, 2019), it seems that this paper uses the exact same dataset. The findings are consistent with prior published results on CCN activation of soot particles; namely that soot with a higher fraction of organic carbon (OC) activates at a shorter oxidation time than soot with less OC. Activation times are strongly dependent on temperature, but not on

relative humidity (up to 75%).

With regards to the data analysis, it seems there is alot of repeat material from the previously published work and this paper, including an entire figure. Is this really necessary? I realize that some overview of the method is a good thing, especially with a method so new and not really used before; but even with the extended discussion in the current paper I feel like I need to read the other publication to fully grasp what is happening. So, I recommend shortening and/or removing some of the repeat information and improving the description of activation time.

This paper also applies the experimental results to a global aerosol-climate model (ECHAM6.3-HAM2.3) by altering parameters that determine the aerosol number concentration of CCN-active particles. In all cases, the modeled CCN and CDNC increase, with varying amounts of increase in different locations around the Earth. The increase makes sense, because it seems like the current model does not consider soot aerosol to be CCN-active at all; thus, any parameterization which does make these particles CCN-active should increase the overall number. The spatial pattern of CCN and CDNC increases is explained as larger increases where soot particle loading is high and/or pre-existing CCN concentrations are low, such that competition for water vapor is minimized.

With regards to the modeling results, my remaining question is what is the big-picture conclusion? It seems two activation times were used in the parameters, reflecting the results from two kinds of experiments in the CSTR, but which result is likely closer to truth? What does a 30-100% increase in CDNC mean for long-term climate projections?

Somewhat related to my modeling questions above, one key improvement that needs to be made to the manuscript is relating miniCAST soot to ambient soot. Because you emphasize the global model results at the end of the paper, the question becomes how well does your experimental protocol mimic what would happen in the atmosphere?

One key element to that is how well does your soot match what is found in the atmosphere? Simply stating that the miniCAST has been used for many studies is not enough. Consider the following references, in addition to providing more details from the papers you already cite:

Le, K., Pino, T., Pham, V., Henriksson, J., Török, S., Bengtsson, P. (2019). Raman spectroscopy of mini-CAST soot with various fractions of organic compounds: Structural characterization during heating treatment from 25°C to 1000°C Combustion and Flame 209(), 291-302. https://dx.doi.org/10.1016/j.combustflame.2019.07.037

Moore, R., Ziemba, L., Dutcher, D., Beyersdorf, A., Chan, K., Crumeyrolle, S., Raymond, T., Thornhill, K., Winstead, E., Anderson, B. (2014). Mapping the Operation of the Miniature Combustion Aerosol Standard (Mini-CAST) Soot Generator Aerosol Science and Technology 48(5), 467 479. https://dx.doi.org/10.1080/02786826.2014.890694

Marhaba, I., Ferry, D., Laffon, C., Regier, T., Ouf, F., Parent, P. (2019). Aircraft and MiniCAST soot at the nanoscale Combustion and Flame 204, 278-289. https://dx.doi.org/10.1016/j.combustflame.2019.03.018

Ess, M., Vasilatou, K. (2018). Characterization of a new miniCAST with diffusion flame and premixed flame options: Generation of particles with high EC content in the size range 30 nm to 200 nm Aerosol Science and Technology 53(1), 1-44. https://dx.doi.org/10.1080/02786826.2018.1536818

Overall, the paper needs some more polish before it is ready for publication.

**Specific comments**

Do not duplicate information from the figure captions to the body of the paper. You only need to give descriptions of the figure colors, etc. once. In the body of the paper, focus on presenting what the figures tell us (not how they tell us).

There is also some duplication with regards to the experimental differences between 2016 and 2018. Polish the writing a bit.

Section 5 - How accurate is your SMPS for diameter? Is 3 nm really significant and measurable? I don't believe that it is because at the end of Section 2.1, you mention that the mode diameter of CBW particles is 90 nm after a pre-mix chamber even though you are size-selecting them to be 100 nm. How do they get smaller? I believe you also said that denuding the particles made no difference.

The experiments which vary by a factor of 2 when all else is supposed to be equal seems very interesting and concerning to me. What else could be different besides the temperature? What was the total particle concentration for these two experiments? Only one experiment is shown in Fig 2; would be nice to see the rest in a Supplemental Section. Is there any basis in the literature for just 4 degrees C to cause that big of a change in CCN number concentration?

Figure 5 - is "statistical none-significance" a common term? I am not familiar.

Are the terms "homogeneous ozone oxidation" and "heterogeneous ozone oxidation" commonly used? I've not seen it said this way before. I'm familiar with homogenous and heterogenous nucleation, and I can guess at what these terms mean; but I wonder if there is a better or more precise way of saying these concepts or not.

Section 6.2 - Better describe what it means to be "statistically significant" in your modeling.

Table 3 is not referenced at all in the text.

Figure S1 isn't really discussed.

**Technical corrections**

There are many cases where "since" was used but "because" is the proper word to use.

Do no use "this" or "it" as a noun in a sentence; too vague.

In many places, "particle and VOC free" should be "particle- and VOC-free" for clarity.

Do not start a sentence with a number, e.g. "65 % of ..."

The date of each experiment is irrelevant for anyone not a part of the study; consider naming the experiments A, B, C, etc.

pg 3 line 16 - "The similar results" needs to be reworded

pg 6 line 10 - missing a space in "humiditywere"

pg 11 line 22 - should be "represents"

pg 12 line 12 - missing space in "Table1"; "lower" should be "shorter"

pg 13 line 7 - parenthesis are messed up

pg 14 line 1 - missing comma after "Figure 4"

pg 14 line 4 - missing comma after "case"

pg 14 line 20 - What does "very" theoretical approach mean?

pg 15 line 7 - missing a period

pg 17 lines 27, 30 - some extra commas and periods here

pg 21 line 12 - "less" could be "faster"

pg 21 line 16 - no comma after "Both"

pg 21 line 29 - When is "shortly"?
* * *

---

## Referee Comment (RC2) · Anonymous Referee #2 · 4 Sep 2019

The manuscript reports a laboratory study of CCN activity of soot particles during their atmospheric aging processes and implementation of the lab results into a global aerosol-climate model. The aging processes mainly consider the heterogeneous ozone oxidation at atmospherically relevant condition. The combination of chamber work with an aerosol-climate model is major strength of this study, as it highlights the importance of accurate CCN treatment in a GCM. The paper is well written overall, so I only have some minor comments for the authors to address.

1. The study only considers the heterogeneous ozone oxidation as the aging pathway. However, in the real atmosphere, the formation of SOA or secondary inorganics can be

more complicated than that. Hence, it is questionable how representative the knowledge gained in this study is when we try to parameterize the aging in a global model. More discussions about the caveats of the results are needed here.

2. It is unclear how the soot aging is treated in the standard ECHAM6.3? Do BC particles move from the externally mixed Aitken model to the accumulation mode during the aging, like the other models do (Wang et al., J. Adv. Model. Earth Syst. 10(10), 2514-2526, 2018)? The enhanced aging will also result in a shorter lifetime of BC in the atmosphere. Is this effect considered in the model of this study?

3. By providing additional source of CCN, how does the new parameterization of BC aging in the ECHAM6.3 affect the radiative forcing of aerosol-cloud interactions in that model?

4. Fig. 5. It surprises me that India which emits lots of BC does not exhibit a strong increase in either CCN or CDNC? Also, for central and southern Africa, why CDNC doesn't respond in spite of the significant increases in CCN?

5. The marked changes in the soot optical properties and radiative forcing during the aging processes should also be fully discussed in the introduction part (i.e. Peng et al., Proc. Natl Acad. Sci. USA, 113(16), 4266-4271, 2016).

---

## Author Comment (AC1) · 16 Oct 2019

We thank the reviewer for the positive feedback on our work. We incorporated the reviewer's comments and thereby improved the understandability of our work and made it more accessible to a broader audience. Detailed answers to the individual comments are given below. For clarity, the reviewers' comments are written in black, and our response in red. Texts from the old version of the manuscript are typed in green and texts from the revised manuscript in blue.
* * *
**Reviewer 1**

**General comments:**

This paper discusses the aging of soot particles by ozone oxidation in an environmental chamber. Aging times of up to 12 hours were achieved, and the data analysis follows recently published work by the same author to interpret results from the continuous-flow stirred tank reactor chamber. Speaking of that prior work (Friebel and Mensah, 2019), it seems that this paper uses the exact same dataset. The findings are consistent with prior published results on CCN activation of soot particles; namely that soot with a higher fraction of organic carbon (OC) activates at a shorter oxidation time than soot with less OC. Activation times are strongly dependent on temperature, but not on relative humidity (up to 75%).

The data presented in this manuscript is not the same data as presented in Friebel and Mensah, 2019). The experimental setup is the same but the experimental setpoints are different While the ozone concentration of the experiments presented herein is either 0 or 200 ppb, the ozone concentrations in Friebel & Mensah are 100 and 50 ppb respectively. The aim of our previous work was to introduce the activation time ($t_{act}$) and its applicability to parameterize the CCN activity of aerosol particles. In this publication, we further investigate activation time distribution and to show that $t_{act}$ is a parameter that can be used to compare results from CSTR experiments to results obtained from other aging experiment designs. Within the current work, we apply $t_{act}$ concept to compare the effect of different aging conditions (ozone concentration, humidity, VOC content of the gas phase) on the change in CCN-activity of soot particles.

With regards to the data analysis, it seems there is a lot of repeat material from the previously published work and this paper, including an entire figure. Is this really necessary? I realize that some overview of the method is a good thing, especially with a method so new and not really used before; but even with the extended discussion in the current paper I feel like I need to read the other publication to fully grasp what is happening. So, I recommend shortening and/or removing some of the repeat information and improving the description of activation time.

As the reviewer pointed out, this experimental concept is new and has never been applied before and the extensive repetitions was aimed to support the reader in getting into the $t_{act}$-concept.

Nevertheless, we shortened and restructured the section 3 "data analysis" in accordance with the reviewer's suggestions and removed the figure extracted from Friebel and Mensah (2019). Still, a certain degree of repetition is unavoidable in order to introduce the reader into the basic principles. It now reads as:

[revised manuscript text omitted]

$$\tau_{CSTR} = \frac{V_{CSTR}}{\dot{V}} \qquad (2)$$

This paper also applies the experimental results to a global aerosol-climate model (ECHAM6.3-HAM2.3) by altering parameters that determine the aerosol number concentration of CCN-active particles. In all cases, the modeled CCN and CDNC increase, with varying amounts of increase in different locations around the Earth. The increase makes sense, because it seems like the current model does not consider soot aerosol to be CCN-active at all; thus, any parameterization which does make these particles CCN-active should increase the overall number.

In ECHAM6.3-HAM2.3 soot aerosol particles can become CCN-active by coating with sulfuric acid (this was added in the appendix 8.2.). This mechanism is at work in all simulations.

Section changed from: (P22 L20-21)

So far BC and POM particles are considered to be not CCN-active within ECHAM6.3-HAM2.3, unless they are internally mixed with soluble components such as sulfate.

To: (P21 L24-26)

So far BC and POM particles are considered to be not CCN-active within ECHAM6.3-HAM2.3, unless they are internally mixed with soluble components such as sulfate. Particles are transferred from the insoluble to the soluble mode when sufficient sulfuric acid gas condenses on insoluble particles to form a monolayer of coating, or by coagulation with soluble particles.

The spatial pattern of CCN and CDNC increases is explained as larger increases where soot particle loading is high and/or
pre-existing CCN concentrations are low, such that competition for water vapor is minimized.

With regards to the modeling results, my remaining question is what is the big-picture conclusion?

The statistically significant changes in CCN concentrations and CDNC show an impact of aging of soot aerosol particles by ozone oxidation on climate relevant variables. The climate impact of this aging pathway, which has not been implemented in global climate models so far, on future climate predictions is an interesting question. However, investigating this aspect demands more detailed knowledge of the reaction kinetics, which cannot be extracted from the data presented herein. The main purpose of the current manuscript is the presentation of the applicability of

1) the CSTR-approach for atmospheric aerosol research,
2) the $t_{act}$-concept for the generation of parameterizations to be implemented in climate models and
3) the experimental results as a basis for future climate predictions.

Within the current manuscript, we focused on the CCN- and CDNC-burden because the CCN activitiy can be extracted from our experimental results directly and this parameter has a direct impact on the CDNC.

It seems two activation times were used in the parameters, reflecting the results from two kinds of experiments in the CSTR, but which result is likely closer to truth?

The two different activation times originate from two different soot types. Non of these soot type is representative for all atmospheric soot particle, because soot an inherently complex material. Depending on its source its composition, morphology etc can vary over a great range. Both soot types are very prominent in the ambient atmosphere as they represent emissions from vehicle and aircraft engines. We therefore believe that both soot types should be part of the evaluation to acquire atmospherically representative result of the global impact of heterogeneous soot ozone oxidation

Following section was added in the manuscript: (P5 L5-8)

The miniCAST has been  used as surrogate for soot emissions from vehicle internal combustion engines (Maricq, 2014; Moore et al., 2014; Mueller et al., 2015) and aircraft engines (Bescond et al., 2014; Durdina et al., 2016). According to Marhaba et al., (2019) high engine thrust levels  can be mimicked by CBK soot, while CBW soot better represents engine emissions at lower thrust levels .

What does a 30-100% increase in CDNC mean for long-term climate projections?

The climate impact of our results is a topic of future scientific research. We encourage climate modelers to implement this aging pathway in their individual models. The results of these different studies will allow for a more holistic evaluation of the climate impact of soot particles.

Somewhat related to my modeling questions above, one key improvement that needs to be made to the manuscript is relating miniCAST soot to ambient soot. Because you emphasize the global model results at the end of the paper, the question becomes how well does your experimental protocol mimic what would happen in the atmosphere?

The CSTR approach allows for aging times of 12 h and beyond. This in turn allows for the experiments to be executed at atmospherically relevant conditions (ozone concentration, particle concentration, relative humidity). Nevertheless, this study focusses on one specific atmospheric aging process and does not mimic all atmospheric aging pahthways (e.g. coating or OH-radical chemistry). By investigation of this isolated aging process we aim to resolve the impact of this specific aging pathway exclusively. However, other aging processes like coating are already implemented in the GCM used herein.

One key element to that is how well does your soot match what is found in the atmosphere? Simply stating that the miniCAST has been used for many studies is not
enough. Consider the following references, in addition to providing more details from
the papers you already cite:

Le, K., Pino, T., Pham, V., Henriksson, J., Török, S., Bengtsson, P. (2019). Raman
spectroscopy of mini-CAST soot with various fractions of organic compounds: Structural
characterization during heating treatment from 25◦C to 1000◦C Combustion and
Flame 209(), 291-302. https://dx.doi.org/10.1016/j.combustflame.2019.07.037

Moore, R., Ziemba, L., Dutcher, D., Beyersdorf, A., Chan, K., Crumeyrolle, S., Raymond, T., Thornhill, K., Winstead, E., Anderson, B. (2014).
Mapping the Operation of the Miniature Combustion Aerosol Standard (MiniCAST) Soot Generator
Aerosol Science and Technology 48(5), 467 479.
https://dx.doi.org/10.1080/02786826.2014.890694

Marhaba, I., Ferry, D., Laffon, C., Regier, T., Ouf, F., Parent, P. (2019). Aircraft and MiniCAST soot at
the nanoscale Combustion and Flame 204, 278-289.
https://dx.doi.org/10.1016/j.combustflame.2019.03.018

Ess, M., Vasilatou, K. (2018). Characterization of a new miniCAST with diffusion
flame and premixed flame options: Generation of particles with high EC content
in the size range 30 nm to 200 nm Aerosol Science and Technology 53(1), 1-44.
https://dx.doi.org/10.1080/02786826.2018.1536818

We thank the reviewer for pointing this out and apologize for the oversight of not having included the appropriate literature to support our statements. We adapted the section where we introduce the soot produced with the miniCAST accordingly.

Section changed from: P4L20-P5L5

Soot particles were produced by a propane fueled Jing Ltd., miniature Combustion Aerosol STandard (miniCAST 4200) generator. Such type of burners and specifically the miniCAST burner have been used widely for the production of soot particles in laboratory studies e.g. (Durdina et al., 2016; Kim et al., 2015; Malmborg et al., 2018; Mamakos et al., 2013; Maricq, 2014; Mueller et al., 2015; Török et al., 2018). The miniCAST was operated in two different settings for the production of soot samples with different organic carbon (OC) contents. The first sample, herein referred to as CAST brown (CBW), with a fuel-air ratio (FAR; $\varphi$) of 1.03 was generated at set-point 6 and characterized by the highest OC content within the range of the burner settings. The second sample, hereafter referred to as CAST black (CBK), with $\varphi$ = 0.95 was generated at set-point 1 and characterized by the lowest OC content within the range of the burner settings. Further details on the miniCAST set-points used during the study are listed in Appendix **Fehler! Verweisquelle konnte nicht gefunden werden.**.

To: (P5 L5-8)

Soot particles were produced by a propane-fueled Jing Ltd., miniature Combustion Aerosol STandard (miniCAST 4200) generator. Such type of burners and specifically the miniCAST burner have been used widely for the production of soot particles in laboratory studies e.g. (Durdina et al., 2016; Kim et al., 2015; Malmborg et al., 2018; Mamakos et al., 2013; Maricq, 2014; Mueller et al., 2015; Török et al., 2018). The miniCAST was operated at two different settings for the generation of soot samples with different organic carbon (OC) contents. The first sample, hereafter referred to as CAST brown (CBW), was generated under fuel-rich conditions (fuel-air ratio; FAR = 1.03). The second sample, hereafter referred to as CAST black (CBK), was generated under fuel-lean condition (FAR = 0.95) . Further details on the miniCAST set-points used during the study are listed in appendix **Fehler! Verweisquelle konnte nicht gefunden werden.**.

The miniCAST has been  used as surrogate for soot emissions from vehicle internal combustion engines (Maricq, 2014; Moore et al., 2014; Mueller et al., 2015) and aircraft engines (Bescond et al., 2014; Durdina et al., 2016). According to Marhaba et al., (2019) high engine thrust levels  can be mimicked by CBK soot, while CBW soot better represents engine emissions at lower thrust levels .

However, we refrain from including Ess and Vasilatou (2018) and Le et. al (2019)  because these publications are based on a different type of miniCAST burner and do not compare the miniCAST soot with other soot, respectively.

Overall, the paper needs some more polish before it is ready for publication.

**Specific comments:**
Do not duplicate information from the figure captions to the body of the paper. You only need to give descriptions of the figure colors, etc. once. In the body of the paper, focus on presenting what the figures tell us (not how they tell us).

We would like to mention that there are different conventions on how to present the content of figures in different scientific communities. Nevertheless, we reduced the duplication of the figure captions in the main text. Due to the novelty of the $t_{act}$ concept, the reader might be unacquainted with the content of the figures and the shape of the curves. We prefer to account for this aspect by continuing to include some of the figure caption content in the main text.

There is also some duplication with regards to the experimental differences between 2016 and 2018. Polish the writing a bit.

In accordance with the reviewers comment we removed the details concerning the experimental differences in section 4.2 "CAST black" and instead refer to the section 2.5 "Experimental procedure and experimental conditions" and 3.4 "Particle losses" for further details.

Section 5 - How accurate is your SMPS for diameter? Is 3 nm really significant and measurable?
The detection of diameter difference of less than 1 nm can be achieved if the initial aerosol particles are size-selected and therefore have a narrow size-distribution. For example Fendel et al. (1995) present resolutions of down to 0.3 nm if the appropriate analysis of the data is performed.

I don't believe that it is because at the end of Section 2.1, you mention that the mode diameter of CBW particles is 90 nm after a pre-mix chamber even though you are size-selecting them to be 100 nm. How do they get smaller?

We apologize for the missing clarity in our text. The aerosol particles that are emitted by the miniCAST are polydisperse and have a broad size distribution peaking at 60 nm This initial particles are allowd to coagulate in the premixing chamber causing the peak diameter to increase to 90 nm thereby increasing the amount of particles with a diameter of 100 nm and beyond. This coagulation step is necessary to allow for a sufficiently high number concentration of particles at 100 nm. The finale particle size selection is performed on these coagulated particles.

I believe you also
said that denuding the particles made no difference.

Denuding the particles had no influence on the CCN activity of the particles. The impact of denuding on the particle diameter could not be determined in our experiments as the size-selecting was performed downstream the denuder. We anticipate a negligible impact on the particle size as the denuder was unheated, and therefore mainly VOC from the gas phase should have been removed

The experiments which vary by a factor of 2 when all else is supposed to be equal
seems very interesting and concerning to me.

We agree that a factor of 2 variation in the results is very interesting and demands further investigation in the future. As discussed in the manuscript, this indicated that the reaction temperature has a major impact on the aging speed.
We would like to highlight the fact, that the uncertainties reported for other aging setups are significantly larger than a factor of 2. For example, the instrumental uncertainty of OFR-studies is reported to be in the range of a factor 5 (Lambe et al., 2011; Simonen et al., 2017) while our instrumental uncertainties are down to $t_{act} \pm 12$min (equivalent to a factor of 0.1). It is a strength of the CSTR approach to allow for the resolution of such day to day variations, which would not be possible with other setups.

What else could be different besides the temperature?
We could not identify another parameter besides the temperature that could have caused the deviation discussed above. But as mentioned before, we believe tailored experiments are needed in the future to resolve the impact of reaction temperature more clearly.

What was the total particle concentration for these two experiments?
While the particle concentration was stable in the feed-in flow throughout the entire duration of an individual experiment (variations on the order $\pm$ 50 #/cc), it ranged from 1000 to 1500 #/cc on the individual days.

Only one experiment is shown in Fig 2; would be nice to see the rest in a Supplemental
Section.
As $t_{act}$ cannot be retrieved directly from the data presented in the diagrams, we implemented Fig. 2 for illustrational purposes only.  Since, all other experiments show the same evolution of the curves and also do not allow to retrieve data points directly we refrain from putting them into the supplement and focus on the relevant data points instead. The raw data, processed data and data published in the manuscript is made available online and can be downloaded from http://dx.doi.org/10.5281/zenodo.2541937 and http://dx.doi.org/10.5281/zenodo.3452036 .

Is there any basis in the literature for just 4 degrees C to cause that big of a
change in CCN number concentration?
Chemical reactions are rather sensitive toward the reaction temperature. However, this aspect is rarely considered in aerosol aging studies since there are so many parameters impacting the reactions significantly. In addition, many experimental setups for aerosol aging (e.g. PAM camber) do not allow for direct temperature control.

The temperature sensitivity of heterogeneous ozone oxidation has been confirmed in model simulations and experimental studies investigating polycyclic aromatic hydrocarbons (PAH) and organic molecules with C=C double bonds. Both chemical structures can be considered as soot surrogates Activation energies of 40 to 80 kJ·mol$^{-1}$ (Berkemeier et al., 2016; Lee et al., 2009; Pöschl et al., 2001; Stephens et al., 1989) have been determined from such experiments. Implementaiton of these activation energies into the Arrhenius law allows for the determination of the relative change in the reaction rates due to a temperature change. Using the activation energy range mentioned above, it can be shown that a change in $t_{act}$ by a factor of 2 can be attributed to the temperature variation reported herein.

Also, the CCN-number concentration does not directly correlate with the temperature. The measured/absolute CCN-number concentration (and activated fraction) is a function of the $t_{act}$ and of the particle liftetime inside the CSTR ($\tau_{age}$). Both parameters are temperature dependent. $\tau_{age}$ depends on the volumetric flowrate and on the particle loss rate and is therefore a function of the temperature as well.

Figure 5 - is "statistical none-significance" a common term? I am not familiar.
This has been changed to: "statistically not significant differences".

Are the terms "homogeneous ozone oxidation" and "heterogeneous ozone oxidation" commonly used? I've not seen it said this way before. I'm familiar with homogenous and heterogenous nucleation, and I can guess at what these terms mean; but I wonder if there is a better or more precise way of saying these concepts or not.

In the chemistry community, "homogenous" and "heterogeneous" reaction are terms used to clarify in which physical phase the individual reaction partners are while the chemical reaction occurs. "Homogeneous" refers to both/all reaction partners being in the same physical state (e.g. gas with gas / ozone and VOCs) while "heterogeneous" refers to the reaction partners to be in different physical states (e.g. gas with solids / ozone with soot)

Section 6.2 - Better describe what it means to be "statistically significant" in your modeling.

Statistical significance is computed from annual mean values of the experiments for each model grid box. This has been added to the text in appendix 8.2. (P18L15-P19L1): "A two-tailed unpaired Student's t-test is computed for each model grid box from annual mean values (20 for each experiment)."

Table 3 is not referenced at all in the text.
A reference to Table 3 has been inserted in the text at P20L5:
The section now reads as: P20L2-7
The largest changes in CDNC occur below about 700 hPa, i.e. for low-level clouds (Figure S1). Again, the impact at $t_{act,ref}$ = 10 h is much more pronounced than at $t_{act,ref}$ = 50 h with a global mean CDNC burden of 3.8 × 10$^{10}$ m$^{-2}$ (+ 17.8 % compared to REF) and 3.5 × 10$^{10}$ m$^{-2}$ (+ 8.9 % compared to REF), respectively (**Fehler! Verweisquelle konnte nicht gefunden werden.**). The largest increases in liquid cloud droplets occur around 60 °N over Europe, Asia and North America causing almost a doubling (+ 93.0 %) in the case of $t_{act,ref}$ = 10 h and an increase by more than 30 % in the case of $t_{act,ref}$ = 50 h.

Figure S1 isn't really discussed.
Figure S1 shows that the largest differences in CCN occur in regions where the CCN concentrations are relatively low in the REF simulation. Following statement was added: (P20L2-3)
The largest changes in CDNC occur below about 700 hPa, i.e. for low-level clouds (Figure S1). Further, Figure S1 shows that mainly low-level clouds are impacted by the additional soot CCN

Following statement was added: (P19L22-24)
As a result of this competition the annual mean values of CCN show the largest differences to the REF experiment in regions where the emissions of BC and POM are large (not shown) and CCN concentrations are relatively low in the REF simulation (see **Fehler! Verweisquelle konnte nicht gefunden werden.** and S1 in the supplement).

**Technical corrections**
There are many cases where "since" was used but "because" is the proper word to use.
Do no use "this" or "it" as a noun in a sentence; too vague..
In many places, "particle and VOC free" should be "particle- and VOC-free" for clarity.
Do not start a sentence with a number, e.g. "65 % of ..."
We thank the reviewer for these comments and polished the writing accordingly.

The date of each experiment is irrelevant for anyone not a part of the study; consider naming the experiments A, B, C, etc.

We do agree with the reviewer from a general perspective as the data reported were recorded in a controlled lab environment and not during e.g. field measurement. However, in this specific case we prefer to keep the date of the experiment as additional parameter. This study contains data from two measurement campaigns that were conducted with a 2 years break in between. Especially for the summer 2016 campaign, we expect that the hot weather had a significant influence on the activated fractions determined, leading to an overall uncertainty in the results of factor of 2 as discussed above.
We improved the references to the individual experiments by mentioning the experiment number in parallel to the date of the experiment. In order to avoid confusing with the labelling of the experimental phases as presented in Figure 2 we decided not to use numbers instead of letters.

pg 3 line 16 - "The similar results" needs to be reworded
done
pg 6 line 10 - missing a space in "humiditywere"
done
pg 11 line 22 - should be "represents"
done
pg 12 line 12 - missing space in "Table1"; "lower" should be "shorter"
done
pg 13 line 7 - parenthesis are messed up
done
pg 14 line 1 - missing comma after "Figure 4"
done
pg 14 line 4 - missing comma after "case"
done
pg 14 line 20 - What does "very" theoretical approach mean?
removed "very"
pg 15 line 7 - missing a period
done
pg 17 lines 27, 30 - some extra commas and periods here
done
pg 21 line 12 - "less" could be "faster"
To keep the wording consistent throughout the manuscript we prefer "less" aging time.
pg 21 line 16 - no comma after "Both"
done
pg 21 line 29 - When is "shortly"?

Data is uploaded now and the data availability statement was updated. P20L32-34

The data presented in this publication can be downloaded from
http://dx.doi.org/10.5281/zenodo.2541937 . The scripts to produce Figures 4 and S1 can be
downloaded from http://dx.doi.org/10.5281/zenodo.3452036 .

**Reviewer 2**

**General comments:**
The manuscript reports a laboratory study of CCN activity of soot particles during
their atmospheric aging processes and implementation of the lab results into a global
aerosol-climate model. The aging processes mainly consider the heterogeneous
ozone oxidation at atmospherically relevant condition. The combination of chamber
work with an aerosol-climate model is major strength of this study, as it highlights the
importance of accurate CCN treatment in a GCM. The paper is well written overall, so
I only have some minor comments for the authors to address.

1. The study only considers the heterogeneous ozone oxidation as the aging pathway. However, in
the real atmosphere, the formation of SOA or secondary inorganics can be more complicated than
that. Hence, it is questionable how representative the knowledge gained in this study is when we try
to parameterize the aging in a global model. More discussions about the caveats of the results are
needed here.

We agree to the reviewer that the oxidation of soot with ozone is only one out of many atmospheric
aging processes. Several of these processes are already implemented in the global climate model
(GCM) used within this manuscript. These processes include the prescribed SOA formation from
gaseous precursors and the condensation of these aging products on externally and internally mixed
Aitken mode particles. Further processes such as coating with sulfuric acid is implemented by
dynamically adjusting the particle properties like the hygroscopicity . The purpose of the study
presented herein is to evaluate the potential global impact of this very specific additional pathway
that competes with aging pathways already implemented.

The impact of different chemical aging pathways on BC in GCMs has been studied by e.g. He et al.
(2016), Huang et al. (2013) and Croft et al. (2005). He et al. (2016) find the largest impact on the
aging rate (= change in CCN-activity) of BC by ozone oxidation at high latitudes (> 60°N and >60 °S)
and above 900 hPa. This is very similar to our results where we find the largest impact in terms of
CCN burden and CDNC burden because of the heterogeneous ozone oxidation north of 60 °.

ECHAM6-HAM uses a simplified treatment of SOA by prescribing the fraction of emitted gaseous
precursors that are transferred to their aging products of lower volatility as well as the fraction of
these product condensing onto preexisting particles.

In contrast to that, He et al. (2016) explicitly account for SOA condensation in their simulations. Still,
they find that ozone aging contributes more than 50 % to the total aging rate of BC in the regions
mentioned above (Fig. 10 therein). Therefore, it can be assumed that accounting for SOA
condensation explicitly in our simulations would lead to qualitatively similar (but quantitatively
different) results as presented in our manuscript. Further, we would like to highlight the fact that the
exceptionally long atmospheric lifetime of soot particles compared to other aerosol particle species

allows for the heterogeneous oxidation to proceed to such a degree that the particles can become CCN-active.

Following section was added:SECTION NOW READS as

Please note that the treatment of secondary organic aerosol (SOA) is simplified in ECHAM6.3-HAM2.3. During emission the soluble and insoluble fractions of SOA are assumed to condense immediately on the soluble Aitken and accumulation and insoluble Aitken mode respectively. However, He et al. (2016) accounted in their experiments for BC aging by condensation of SOA and found that their chemical aging mechanism still accounted for more than 50% of the BC aging rate at high latitudes (polewards of 60°N/S) and above 900 hPa.

2. It is unclear how the soot aging is treated in the standard ECHAM6.3? Do BC particles move from the externally mixed Aitken model to the accumulation mode during the aging, like the other models do (Wang et al., J. Adv. Model. Earth Syst. 10(10),2514-2526, 2018)?

In the standard ECHAM6.3-HAM2.3 GCM the BC particles move from the externally mixed Aitken mode into the internally mixed Aitken mode by coating with sulfuric acid. Thereby they become CCN-active. This additional information was added in the appendix P21L24-26.

So far BC and POM particles are considered to be not CCN-active within ECHAM6.3-HAM2.3, unless they are internally mixed with soluble components such as sulfate. Particles are transferred from the insoluble to the soluble mode when sufficient sulfuric acid gas condenses on insoluble particles to form a monolayer of coating, or by coagulation with soluble particles.

The enhanced aging will also result in a shorter lifetime of BC in the atmosphere. Is this effect considered in the model of this study?

Following statement was added in the manuscript which should answer the reviewers question. P17L10-13

Please note that the increase of the soot particle's hygroscopicity due to oxidation with $O_3$ increases the soot particle removal rate from the atmosphere, e.g. due to a higher wet-deposition rate. As a result, the average atmospheric lifetime of soot particles decreases. However, the reduction of the soot particle lifetime was below 2% in both scenarios. Since this lifetime reduction is statistically not significant, its impact on the CCN burden and CDNC was not considered within this study.

3. By providing additional source of CCN, how does the new parameterization of BC aging in the ECHAM6.3 affect the radiative forcing of aerosol-cloud interactions in that model?

The focus of the study presented in the manuscript is to proof the applicability of the CSTR-approach for the investigation of changes in the CCN-activity, to show that the $t_{act}$-concept can be implemented in climate models and to show that the experimental results are potentially relevant for future climate predictions. We only published data concerning the CCN concentration and CDNC including their spatial patterns because these are the variables which are directly affected by this new aging pathway. The influence of this so far unconsidered aging pathway onto radiative forcing of aerosol-cloud interactions is an interesting question. However, investigating this does not lie within the scope of this study.

4. Fig. 5. It surprises me that India which emits lots of BC does not exhibit a strong increase in either CCN or CDNC?

The highest BC emission in India can be found in Northern India, especially in the densely populated Ganges river valley. This is the same region where the CCN burden is one the highest worldwide, as can be seen in Figure 4a (REF). In the $t_{act,ref}$ = 10h scenario (Fig 4c) the absolute increase of the CCN burden matches the increase in the northern mid-latitudes. In the $t_{act,ref}$ = 50h scenario (Fig 4e) the increase in the CCN burden is equally strong as in central Africa.

See below for the change in the CDNC burden

Also, for central and southern Africa, why CDN
doesn't respond in spite of the significant increases in CCN?

There is a small but statistically significant increase in CDNC in central Africa and parts of southern Africa. This response is weaker than in the northern mid latitudes, because stratiform clouds are less frequent in central or southern Africa than over land in the Northern Hemisphere mid latitudes. Therefore, the changes in annual mean CDNC are smaller in these regions.

The section in the manuscript was change from: P20L29-32

Similar to the changes in CCN burden, the strongest increases in CDNC can be found over land in the Northern Hemisphere with lower values in the tropics. Likewise, the area where CDNC changes occur is smaller in the tropics than at higher latitudes in the Northern Hemisphere. This is because stratiform liquid clouds occur more often in mid latitudes than in the tropics, which is indicated by the higher CDNC burden in mid latitudes (**Fehler! Verweisquelle konnte nicht gefunden werden.**b).

To: P19L29-P2L3

Similar to the changes in CCN burden, the strongest increases in CDNC can be found over land on the Northern Hemisphere. However, over the tropics the increase in CDNC is much lower than the increase in the CCN burden. The difference is caused by the higher abundance of stratiform liquid clouds in mid latitudes compared to the tropics, which is indicated by the higher CDNC burden in mid latitudes (Figure 4b). Please note that ECHAM6.3-HAM2.3 accounts for CDNC from detrained cloud water of convective clouds but otherwise the convective cloud parameterization does not consider CDNC (Neubauer et al., 2019). Therefore, our simulations could underestimate the impact of heterogeneous ozone oxidation of soot particles on CDNC, in particular where convective clouds are common like the tropics. The largest changes in CDNC occur below about 700 hPa, i.e. for low-level clouds (Figure S1).

5. The marked changes in the soot optical properties and radiative forcing during the aging processes should also be fully discussed in the introduction part (i.e. Peng et al., Proc. Natl Acad. Sci. USA, 113(16), 4266-4271, 2016)

Unfortunately, we had no opportunity to measure the optical properties of the particles prior to and throughout the aging process. Therefore, we refrained from changing the scheme how the direct aerosol effect is implemented in ECHAM6.3-HAM2.3.

The same applies to radiative effects due to aerosol-cloud interactions. During the experimental part of the study, we focused on the change in CCN-activity. In the modeling part, we therefore only focused on the parameters that are connected directly to the change in CCN activity of the soot particles. As the optical properties of soot and the radiative forcing are not part of this study we do not introduce them in great detail.